# Genetic screens in isogenic mammalian cell lines without single cell cloning

Peter C. DeWeirdt[1,2], Annabel K. Sangree[1,2], Ruth E. Hanna[1,2], Kendall R. Sanson[1,2], Mudra Hegde [1], Christine Strand [1], Nicole S. Persky[1] & John G. Doench [1]*

Isogenic pairs of cell lines, which differ by a single genetic modification, are powerful tools for understanding gene function. Generating such pairs of mammalian cells, however, is labor-intensive, time-consuming, and, in some cell types, essentially impossible. Here, we present an approach to create isogenic pairs of cells that avoids single cell cloning, and screen these pairs with genome-wide CRISPR-Cas9 libraries to generate genetic interaction maps. We query the anti-apoptotic genes *BCL2L1* and *MCL1*, and the DNA damage repair gene *PARP1*, identifying both expected and uncharacterized buffering and synthetic lethal interactions. Additionally, we compare acute CRISPR-based knockout, single cell clones, and small-molecule inhibition. We observe that, while the approaches provide largely overlapping information, differences emerge, highlighting an important consideration when employing genetic screens to identify and characterize potential drug targets. We anticipate that this methodology will be broadly useful to comprehensively study gene function across many contexts.

---

[1] Genetic Perturbation Platform, Broad Institute of MIT and Harvard, 75 Ames Street, Cambridge, MA 02142, USA. [2] These authors contributed equally: Peter C. DeWeirdt, Annabel K. Sangree, Ruth E. Hanna, Kendall R. Sanson. *email: jdoench@broadinstitute.org

Genetic interaction networks can reveal unexpected connections between genes and suggest the functions of uncharacterized genes, which may prove critical for interpreting genetic signal from genome-wide association studies of common disease states. Although yeast knockout crosses have yielded rich genetic interaction networks[1–4], in mammalian cells, constructing such networks is orders of magnitude more complicated, due to increased genome size, diversity of cell types, and numerous technical factors. RNAi[5] and CRISPR technology[6–10] have been used to study pairwise interactions for up to hundreds of genes[11]; however, screening all combinations of protein coding genes in the human genome would require, at bare minimum, approximately 400 million perturbations and 200 billion cells, which is equivalent to 5000 concurrent genome-wide screens. This scale is exacerbated by the diversity of cell types in which to study such interactions.

A second, complementary approach to query genetic interactions leverages isogenic pairs of human cells. Initial gene knockout techniques in human cell lines have yielded valuable insights but were quite laborious to execute[12–18]. Recently, CRISPR technology has enabled cell line engineering for a broader range of researchers, but that is distinct from making it easy. Programming the Cas9 target site is as simple as ordering a short nucleic acid, in contrast to the more cumbersome task of assembling a customized pair of zinc-finger nucleases or TALENs[19]. After design of the targeted nuclease, however, substantial work remains: isolation of single cells, often across multiple 96-well plates; expansion for several weeks while colonies form; isolation of genomic DNA from replicated plates; and finally, PCR, sequencing, and analysis to determine which colonies have the intended genotype[20]. Indeed, off-the-shelf knockout clones, which are available in only a very limited number of cell lines, can be purchased from vendors for thousands of dollars, and the customized generation of a knockout clone in a cell line of interest costs tens of thousands of dollars. Thus, there is a great need for approaches that obviate the need to generate single-cell clones and enable the creation of large-scale genetic interaction maps for genes of interest in relevant cell types.

Here, we leverage orthogonal Cas enzymes from *S. pyogenes* and *S. aureus* to conduct genome-wide CRISPR screens in paired mutant cell lines without the need for single-cell cloning; we call this approach "anchor screening", as the single genetic mutant anchors the resulting interaction network. We selected *BCL2L1*, *MCL1*, and *PARP1* as anchor genes, as they each have well-established genetic interactions to facilitate benchmarking. They are also the subject of intense clinical development, allowing for both a comparison between small-molecule inhibition and genetic knockout, and for PARP inhibitors, potentially an expansion of the genotypes beyond *BRCA1* and *BRCA2* mutant tumors in which these drugs may show efficacy. The rich set of resulting genetic interactions shown here coupled with the ease of conducting such screens illustrate the power of this technology.

## Results

**Anchor screening rationale.** Genetic screens with CRISPR technology often start with the creation of a cell line stably expressing Cas9, integrated into the genome via lentivirus or piggybac transposase[21,22]. Because only a single element is delivered, this can be performed at small scale, and the resulting cells expanded over the course of several weeks to the tens of millions of cells required for genome-scale libraries of single-guide RNAs (sgRNAs, hereafter referred to as "guides"). In theory, one could also introduce a guide targeting a gene of interest at this step, to create a pool of knockout cells, and subsequently screen that population of cells against a library of guides.

However, if there is any selective pressure against the knockout cells, they will become underrepresented during scale-up (Supplementary Fig. 1). For example, assume that (i) unmodified cells, or those with in-frame indels, double every 24 h, and (ii) knockout cells represent 90% of the pool at the start. If the knockout cells have a 20% slower growth rate, they will represent less than half of the population after 3 weeks of proliferation. Inducible CRISPR systems could be helpful, but all of them require the use of additional components, such as recombinases, degrons, dimerization domains, transcriptional activators, or transcriptional repressors, as well as small-molecule inducers, many of which have biological effects. Further, recent comparisons have shown that current systems often have substantially less activity than constitutive versions, or demonstrate leakiness; additionally, performance is typically cell-type dependent[23,24]. Thus, there is a need for a simple method to generate cells poised for gene editing, expand them with no selective pressure, and trigger efficient knockout only when ready to begin a genetic screen.

Previously, we and others developed *S. aureus* Cas9 (SaurCas9) for screening applications and paired it with *S. pyogenes* Cas9 (SpyoCas9) to enable combinatorial screens of "some-by-some" genes[9,25]. Small modifications to the vector designs enable us to perform "one-by-all" screens with a workflow identical to standard genome-wide screens. The first vector, deemed the anchor vector, delivers SpyoCas9 and a guide compatible with *S. aureus* Cas9 (Saur-guide); the second vector delivers SaurCas9 and a guide cassette compatible with *S. pyogenes* Cas9 (Spyo-guide), which delivers the library of choice (Fig. 1a). Thus, a guide can be cloned into the anchor vector, delivered at small scale, and the resulting population of cells expanded. Critically, because the guide is paired with the wrong Cas9, no editing will occur and thus there is no selective pressure during cell expansion. Finally, the library is introduced, and each cell will generate approximately simultaneous knockout of both the anchor gene and the gene targeted by the library (Fig. 1b). This process can be completed in ~5 weeks, less time than is required to generate and validate single-cell clones, let alone screen them.

**Anchor screens for the anti-apoptotic genes *BCL2L1* and *MCL1*.** We selected two genes as anchors to test this approach, the anti-apoptotic genes *MCL1* and *BCL2L1*, which themselves are a well-validated synthetic lethal pair[8,9,26]. For each gene we selected a previously validated Saur-guide[9] for use in the anchor vector and generated stable populations in the Meljuso melanoma cell line and the OVCAR8 ovarian cancer cell line; we used the empty anchor vector to generate a control population. We introduced the Brunello genome-wide library into the library vector, which has four guides per gene and ~78,000 total guides[27]. We transduced the library vector into the resulting six anchor lines in duplicate, selected with puromycin for 5–7 days, and maintained the population with at least 500× coverage for an additional 2 weeks. As an additional experimental arm, we treated the control cells with either A-1331852 or S63845, small-molecule inhibitors of BCL2L1 (ref. [28]) or MCL1 (ref. [29]), respectively, for the final 2 weeks of the experiment (Fig. 1b). At the end of the screen, we pelleted cells, prepared genomic DNA, retrieved the library guides by PCR, and performed Illumina sequencing to determine the abundance of each guide.

To detect genetic interactions with the anchor gene, we first calculated the log2-fold-change (LFC) compared to the initial library abundance, as determined by sequencing the plasmid DNA (Supplementary Data 1) and observed that replicates were well correlated (Supplementary Table 1). For each anchor cell line, we compared LFC values to the corresponding control cell

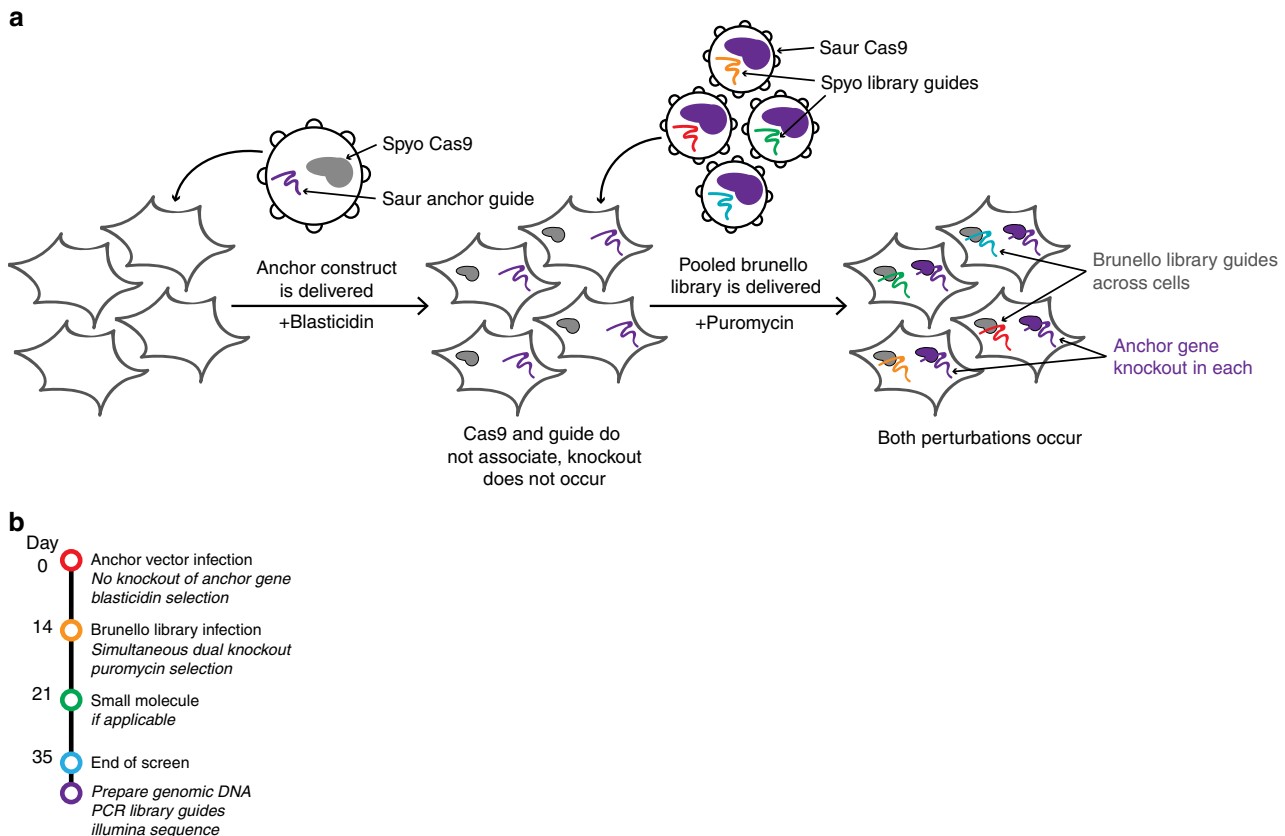

**Fig. 1 Development of isogenic cell lines and the anchor screening approach using a two-Cas9 system. a** Schematic of anchor screens performed with the Brunello library. Saur components in purple and Spyo in remaining colors. **b** Timeline by which the screens were executed.

line by fitting a nonlinear function and calculating the residual for each guide; a positive residual represents a buffering interaction, and a negative residual represents a synthetic lethal interaction (Fig. 2a). Residuals for individual guides were combined at the gene level by calculating a Z-score (Fig. 2b); the same approach was used to determine sensitivity and resistance genes for the small molecules. The gene-level results from all screens are available in Supplementary Data 2.

When anchoring on *BCL2L1* knockout (Fig. 2c), *MCL1* scored strongly in both Meljuso (ascending rank 2, Z-score −9.6) and OVCAR8 (ascending rank 2, Z-score −6.1). Conversely, when anchoring on *MCL1* (Fig. 2d), *BCL2L1* emerged as a top synthetic lethal interaction in both Meljuso (ascending rank 5, Z-score −5.0) and OVCAR8 (ascending rank 1, Z-score −7.3). These relationships were also captured by the small-molecule screens. With the BCL2L1 inhibitor A-1331852, *MCL1* was a top sensitizer gene in both Meljuso (ascending rank 2, Z-score −15.9) and OVCAR8 (ascending rank 2, Z-score −7.8). Likewise, when screened with the MCL1 inhibitor S63845, *BCL2L1* scored strongly in both Meljuso (ascending rank 2, Z-score −9.7) and OVCAR8 (ascending rank 1, Z-score −5.9). Thus, these genome-wide anchor screens identified the expected synthetic lethal relationship between these genes, which were also observed with small-molecule inhibition.

Other genes with well-established roles in apoptosis scored in these screens. Previously, we reported synthetic lethality with *BCL2L1* and *BCL2L2* (ref. [9]), which was confirmed in these genome-wide screens: in the *BCL2L1* anchor screen, *BCL2L2* scored highly in Meljuso (ascending rank 4, Z-score −6.8) and OVCAR8 (ascending rank 3, Z-score −5.2). Additionally, *BCL2* scored strongly in Meljuso (ascending rank 3, Z-score −7.9) but weakly in OVCAR8 (ascending rank 926, Z-score −1.9),

a cell-type difference observed previously[9]. Further, we saw buffering interactions between *BCL2L1* and the pro-apoptotic genes *TP53* (average descending rank 1, Z-score 5.4), *BAX* (descending rank 2, Z-score 4.2), and *PMAIP1* (also known as *NOXA*, descending rank 3, Z-score 3.7). These genes were also the top three resistance hits for the small-molecule A-1331852.

Additional genes emerged as strong hits. The E3 ubiquitin ligase *MARCH5* showed strong synthetic lethality with *BCL2L1* in both Meljuso (ascending rank 1, Z-score -11.3) and OVCAR8 (ascending rank 1, Z-score -6.1). Previous studies have shown that *MCL1* levels are elevated in *MARCH5* knockout cells[30], and siRNA-mediated knockdown of *MARCH5* led to loss of *MCL1*-mediated resistance to the BCL2-family inhibitor ABT-737[31]. Two additional synthetic lethal hits with *BCL2L1* are the E2 ligases *UBE2J2* (ascending rank 3, Z-score −7.2 across all conditions) and *UBE2K* (ascending rank 5, Z-score −5.1). *WSB2*, a relatively unstudied gene that contains a SOCS box, a domain proposed to recruit ubiquitination factors to bound proteins[32] scored as a top synthetic lethal hit in the *MCL1* anchor screens in both Meljuso (ascending rank 2, Z-score −5.8) and OVCAR8 (rank 16, Z-score −4.7), as well as with the MCL1 inhibitor S63845 (ascending rank 1, Z-score −14.5 in Meljuso; ascending rank 2, Z-score −5.8 in OVCAR8). Nearly all hits were private to *BCL2L1* or *MCL1*, with only *TP53* and *WSB2* scoring across both anchor genes (Fig. 2e). Thus, these screens connected several novel and understudied genes to the intrinsic apoptosis pathway via genetic evidence, and these are worthy of future biochemical study to determine their mechanism.

**Network analyses**. To understand the generalizability of these relationships, we queried the Cancer Dependency Map (DepMap)[33,34], a compendium of genome-wide RNAi and CRISPR screens performed across hundreds of cancer cell lines.

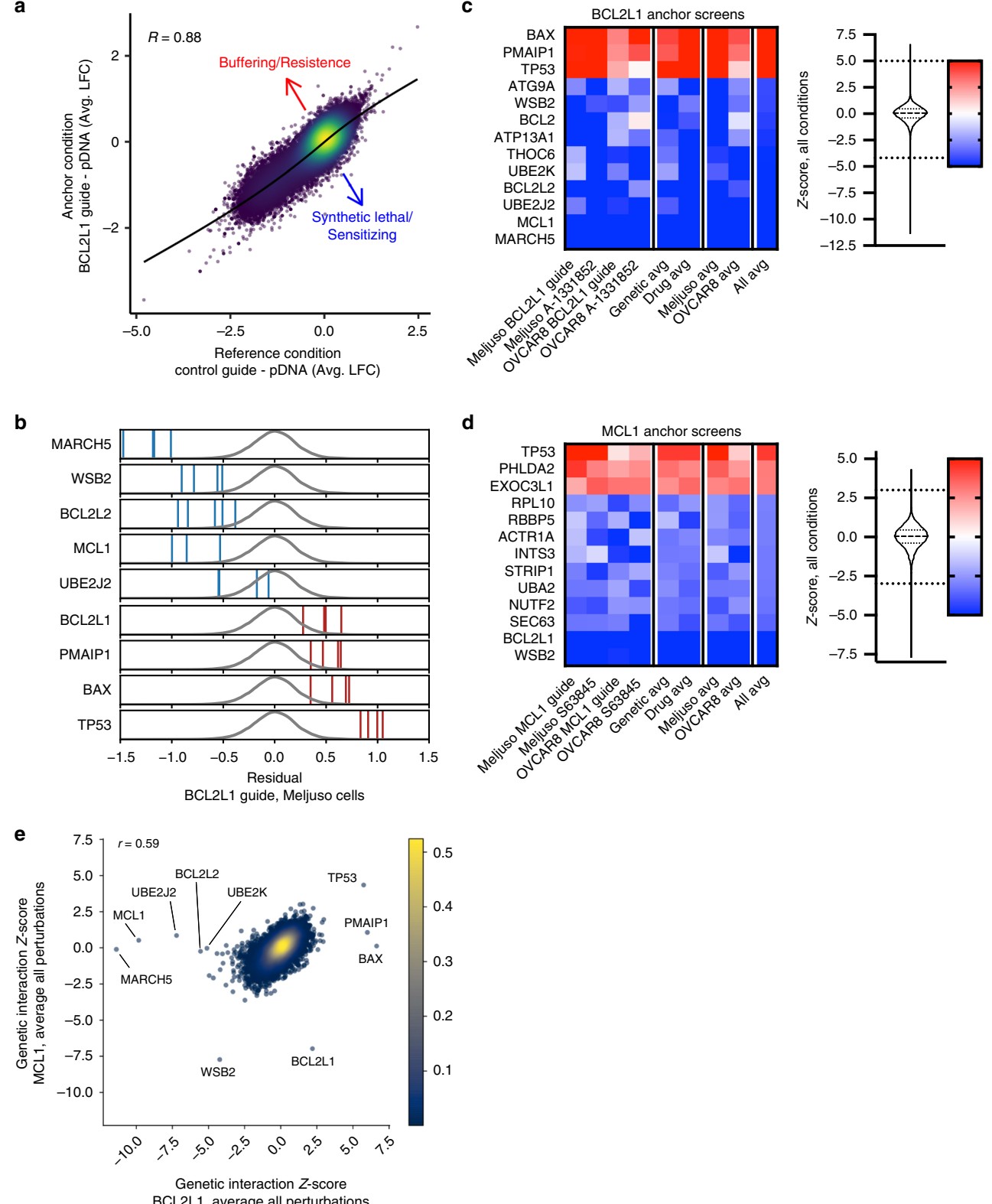

**Fig. 2 Anchor screens of *BCL2L1* and *MCL1* recover known and novel interactions. a** Average log2-fold changes for guides in Meljuso cells for control and *BCL2L1* knockout lines. Points are colored by density. Pearson correlation coefficient is indicated. **b** Residuals for guides from the *BCL2L1* anchor screen in Meljuso. Blue and red lines correspond to lethal and buffering guides respectively. Density of all guides is indicated by the gray distribution. **c** Top 13 hits ranked by absolute average *Z*-score across all *BCL2L1* screens. Color scale of *Z*-scores is shown to the right. A violin plot representing the distribution of *Z*-scores is adjacent to the color scale, along with two dotted lines representing the cutoffs for gene hits shown. The color scale is floored at −5 and ceilinged at 5. **d** Top 13 hits ranked by absolute average *Z*-score across all *MCL1* screens, as in **c**. **e** Comparison of average *Z*-scores for all *MCL1* and *BCL2L1* perturbations screened with the Brunello library. Genes with an absolute average *Z*-score greater than 5 in either condition are labeled.

Here, correlation in fitness effects across cell lines suggests a functional relationship between genes[35–37]. Focusing on the CRISPR data screened with the Avana library, *MARCH5* and *MCL1* show a strong co-dependency ($R = 0.66$); *UBE2J2* ($R = 0.38$) is the second-best correlate of *MARCH5* dependency after *MCL1*, and *UBE2K* ranks fifth ($R = 0.29$). Furthermore, the best correlate to *WSB2* essentiality is *BCL2L2* ($R = 0.39$) and *MCL1* is ranked second ($R = 0.29$). Likewise, in the Project DRIVE RNAi screens[38], the top correlate of *WSB2* co-essentiality is *MCL1* ($R = 0.47$), *BCL2* ranks fourth ($R = 0.39$) and *MARCH5* ranks seventh ($R = 0.39$). Thus, the sensitizers uncovered by the anchor screens are supported by orthogonal large-scale datasets.

Encouraged by these observations, we used a network approach to organize these data further. We selected the top 210 genes (nodes) with an absolute average *Z*-score greater than 2 across all *BCL2L1* screens, and created a network using co-essentiality correlations from the DepMap as edges, with an absolute cutoff of 0.2, which represent 0.45% of all correlations in the dataset (Supplementary Fig. 2). We used a graph-based community detection algorithm[39] to uncover clusters within the co-essentiality network (Supplementary Fig. 3). The clustering revealed densely connected groups of genes, one of which contained 9 of our top 12 hits by absolute average *Z*-score. In this cluster, *MCL1* and *MARCH5* are connected with 10 and 9 genes respectively, making them the two most central hits of the group (Fig. 3a).

We also examined the STRING database[40], which collates gene interactions based on protein–protein interactions, gene ontologies, and other curated annotations. We used a combined score cutoff of 400 to define edges in the STRING network (Supplementary Fig. 4), corresponding to a medium confidence cutoff. We highlight one cluster that contained many of the strongest hits from the genetic screens (Fig. 3b). In both the STRING and DepMap networks we saw an enrichment for edges when compared with random networks of genes of the same order (Supplementary Fig. 5a). Thus, orthogonal network sources reveal a high level of connectivity between the top genes identified by these anchor screens.

Of the top 20 hits in the screen, all of them are connected to at least one other gene in at least one of the two networks, including 5 that are only detected in the DepMap network, showing that co-essentiality can reveal functional relationships that are currently unannotated in the STRING database (Fig. 3c). We performed the same analyses for the hits from the *MCL1* anchor screen (Supplementary Figs. 5b, 6 and 7), and again saw an enrichment for edges compared to random networks with both STRING and DepMap.

**Validation**. To validate top genes, we performed a competition assay in Meljuso cells in which EGFP labels cells with Cas9 and we assess the fraction of EGFP-positive cells over time by flow cytometry (Supplementary Fig. 8a). We observed that loss of *WSB2* or *BCL2L1* sensitizes cells to S63845, whereas loss of *MARCH5* or *MCL1* sensitizes cells to both A-1331852 and navitoclax, a small-molecule inhibitor of BCL2L1 and other BCL2 proteins (Supplementary Fig. 8b).

To further confirm these results, we performed screens using the small-molecule inhibitors in a third cell line, the melanoma line A375, using an orthogonal library of Spyo-guides, Gattinara (Supplementary Data 3). Gattinara is designed with two guides per gene, to reduce the cost of executing these screens, and is complementary to Brunello, in that targeting guides are generally not shared across these libraries (Supplementary Note 1). With the BCL2L1 inhibitor A-1331852 (Supplementary Fig. 9), the top four sensitizer genes were *UBE2J2* (*Z*-score −13.6), *BCL2L2*

(−12.7), *MCL1* (−10.9), and *MARCH5* (−9.8); *UBE2K* ranked tenth (−6.7). Likewise, with the MCL1 inhibitor S63845, *WSB2* and *BCL2L1* scored as the first (*Z*-score −7.7) and third (*Z*-score −6.0) sensitizer hits, respectively, confirming that the strongest genes observed in Meljuso and OVCAR8 cells reproduce in a third cell line with additional guides.

**Spyo-only approach**. Validating Saur-guides likely represents an additional experimental step when considering an anchor screen, as there is comparatively little pre-existing data for this system, whereas many Spyo-guides are already well-vetted. Thus, we tested an anchor screening strategy that uses only SpyoCas9 to generate both knockouts (Fig. 4a). We generated a secondary library targeting 390 potential hit genes that showed evidence of activity in the primary *MCL1* and *BCL2L1* screens as well as 857 non-scoring genes, with 10 guides per gene (Supplementary Data 4). We cloned the library into lentiCRISPRv2 and conducted anchor screens using three Spyo-guides targeting *MCL1* in A375 cells, as well as an S63845 arm. We saw strong correlations across all three guides (Supplementary Fig. 10a–c), suggesting this approach can reliably identify interactors. These secondary screens validated *BCL2L1* and *WSB2* as top synthetic lethal hits with *MCL1* (Fig. 4b). Three members of a cullin-RING ubiquitin ligase complex also scored, *CUL5* (ascending rank 4, *Z*-score −9.79), *RNF7* (ascending rank 5, *Z*-score −8.89), and *UBE2F* (ascending rank 6, *Z*-score −8.01), which are themselves well-correlated in DepMap and were recently identified as modulators of sensitivity to MCL1 inhibitors in lung cancer cells[41]. Only in secondary screens did these three genes score strongly together (Supplementary Fig. 10d–e), highlighting the value of conducting secondary screens with more guides per gene.

**Comparison to single-cell clones**. To evaluate how screening single-cell clones compares with anchor screening, we commissioned the creation of two *MCL1* knockout clones. Notably, one clone was characterized as *MCL1* −/−/−, whereas a second was characterized as *MCL1* −/−/−/−; according to the vendor, there was no indication of a remaining wild-type allele in the former clone. We first characterized these clones by dosing with the BCL2L1 inhibitor A-1331852 (Fig. 4c). The −/−/−/− clone showed far more sensitivity than the −/−/− clone, suggesting that a wild-type allele remained in the −/−/− clone but escaped detection. We then screened both clones with the secondary library and saw that hits were much stronger with the −/−/−/− clone than the −/−/− clone, further supporting this hypothesis (Fig. 4d). These observations underscore the potential difficulty of properly characterizing single-cell clones, especially in cancer cell genomes that can be heterogeneous and unstable.

Across the two clones, top hits were generally consistent, with *BCL2L1* and *WSB2* scoring as strong synthetic lethal hits. However, differences emerged. For example, *DUSP4* scored as a synthetic lethal gene in the −/−/− clone (rank 11, *Z*-score −5.06) but a resistance gene in the −/−/−/− clone (rank 1, *Z*-score 16.10). Similarly, *CUL3* scored as a synthetic lethal gene in the −/−/−/− clone (rank 2, *Z*-score −13.47) but did not have a strong phenotype in the −/−/− clone (*Z*-score 0.96). When we compare the average *Z*-score across all three *MCL1* anchor guides to the −/−/−/− clone, we see the single-cell clone recapitulates top hits from the anchor screens, including *RNF7* (*Z*-score −6.56, rank 11), *UBE2F* (*Z*-score −5.10, rank 28), and *CUL5* (*Z*-score −5.07, rank 29) (Fig. 4e). Considered in reverse, we see that some of the strongest hits from the −/−/−/− clone have a relatively small or no effect in the anchor screens, notably *CUL3* (*Z*-score −2.00, rank 67) and *DUSP4* (*Z*-score −0.87).

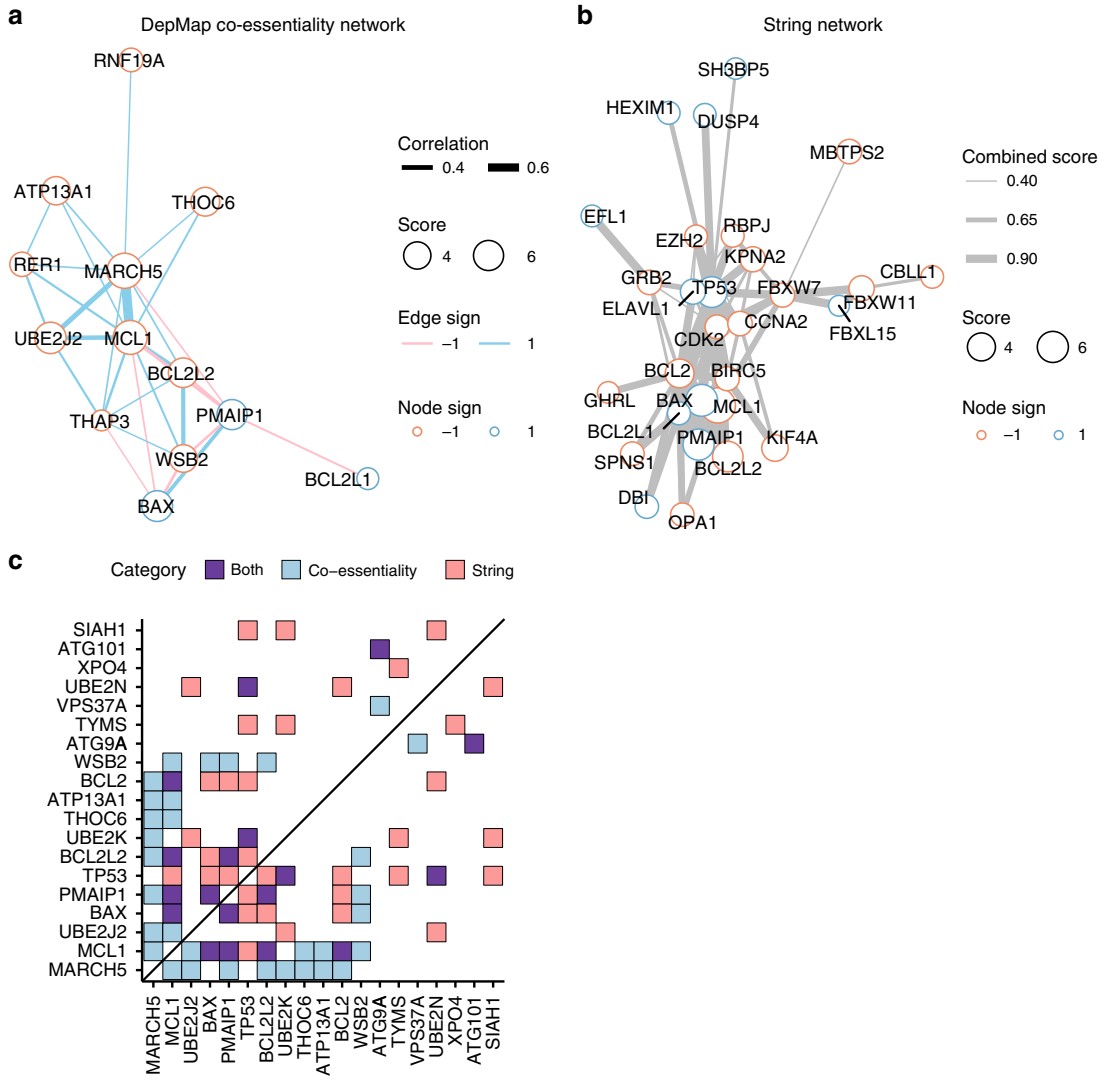

**Fig. 3 BCL2L1 anchor screens reveal functionally coherent clusters of genes. a** Cluster of top hits from the DepMap co-essentiality network. Nodes represent genes and the size of each node is proportional to its average Z-score across all screens. Genes with an absolute average Z-score greater than 2 across BCL2L1 conditions are included in the network. Edges represent Pearson correlations across co-essentiality profiles in DepMap. Edges are drawn between genes with an absolute correlation greater than 0.2. Clustering was done by modularity optimization and a single cluster was chosen for visualization. **b** Cluster of top hits from the STRING network. Nodes are the same as **a**. Edges represent combined score in STRING. Edges are drawn between genes with a STRING combined score greater than 0.4. Clustering was done the same as **a**. **c** Interactions from DepMap and STRING between the top 20 hits by absolute Z-score. Genes are ordered by absolute average Z-score. Edge cutoffs are the same as **a** and **b**.

To compare all three approaches—anchor guide, small molecule, and single-cell clone—we examined the top five hits by absolute Z-score from each screen (Fig. 4f). We saw that six of the eight hits from the anchor screens score strongly (absolute Z-score greater than 5) with S63845 or the −/−/−/− clone, whereas only three of eight hits from the single-cell clones score strongly in at least one orthogonal arm. Importantly, we compared the log2-fold changes for the unmodified A375 cells we obtained from CCLE, which were used for the small molecule and anchor screens, and the unmodified A375 cells the vendor provided along with the MCL1 single-cell clones, and the viability effects were well correlated, indicating that observed differences were not an artifact of phenotypic drift in the reference cell line (Supplementary Fig. 10f). Thus, these differences may be a true biological effect related to the gene dosage of MCL1. However, this may also be an artifact of single-cell cloning, in that each clone contains private mutations or epigenetic alterations. Finally, it is also possible that long-term adaptation to loss of a gene may occur in

a single-cell clone, whereas in anchor screening gene disruption is acute.

Overall, anchor screens with BCL2L1 and MCL1 identified both expected and previously uncharacterized partners, which were supported by parallel small-molecule screens. Further, examination of orthogonal data sources, the STRING and DepMap co-essentiality networks, provides additional confidence in the relevance of these interactions and the validity of this approach. We additionally demonstrate that a SpyoCas9-only anchor screening approach can effectively identify synthetic lethal hits and may be a preferable approach for researchers who have already validated effective Spyo-guides targeting their gene of interest. Finally, we screened two MCL1 knockout single-cell clones, and although top synthetic lethal hits such as BCL2L1 and WSB2 were consistently observed in both, one of the two clones shared fewer top hits in common with either the anchor screens or the small molecule, highlighting the challenge of generalization that may emerge when using clonal cell lines.

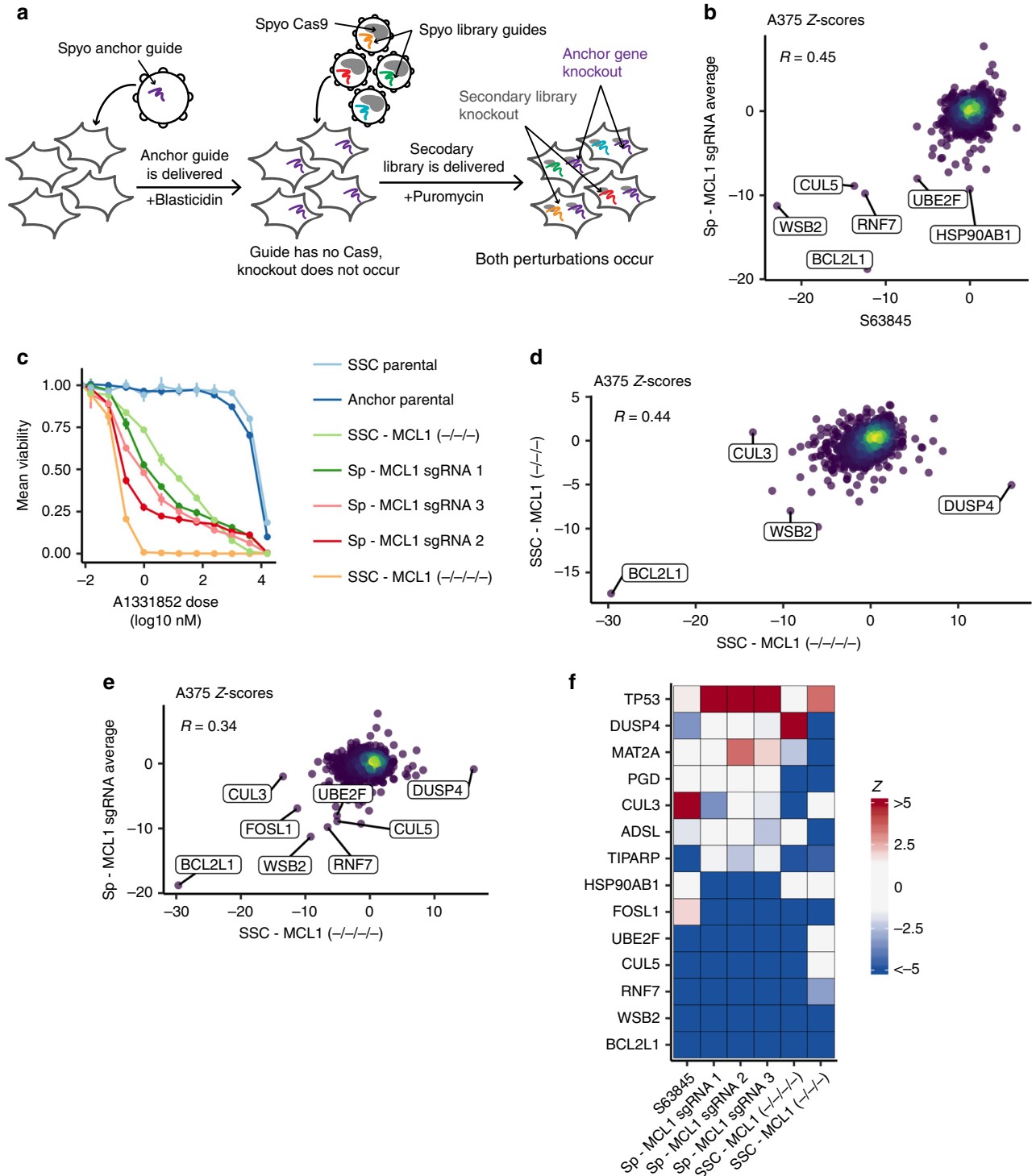

**Fig. 4 *MCL1* secondary screens comparing alternative screening approaches. a** Schematic of Spyo-only approach screened with the secondary library. Spyo anchor perturbation shown in purple, Spyo library perturbations shown in remaining colors. **b** Z-scores for *MCL1* anchors, averaged across three guides compared to S63845 treatment, screened with a secondary library in A375 cells. Points are colored by density. Pearson correlation coefficient is indicated. Top six hits, ranked by average absolute Z-score across *MCL1* guide conditions, are labeled. **c** Dose response curves for cells treated with A-1331852. Cells include two A375 parentals, two A375 *MCL1* knockout clones, and three A375 *MCL1* anchor knockouts. The line range represents one standard deviation for doses where multiple viability measurements were collected. **d** Same as **b** but for the *MCL1* −/−/− knockout clone vs the *MCL1* −/−/−/− knockout clone. Top two hits ranked by absolute average Z-score and top two differences between single-cell clone conditions are labeled. **e** Same as **b** but for *MCL1* anchors, averaged across all guides vs the *MCL1* −/−/−/− knockout clone. Top six and five hits ranked by absolute average Z-score for *MCL1* guide conditions and the *MCL1* −/−/−/− knockout clone, respectively, are labeled. **f** Heatmap of Z-scores for the top five hits from all secondary screens, ranked by absolute Z-score. Genes and screens are ordered by mean Z-score.

**Anchor screens with *PARP1* knockout and PARP inhibitors**. To confirm that this approach can be informative when targeting a gene involved in a different cellular function, we performed anchor screens against the DNA damage response gene *PARP1*. The synthetic lethal interaction between *PARP1* and *BRCA1* is well described[42], and PARP inhibitors are clinically approved for the treatment of tumors with *BRCA* mutations[43]. Identifying additional genetic lesions that also synergize with PARP inhibition would thus be valuable, and could expand the population of patients who may benefit from this therapy. We designed two guides against *PARP1* and conducted anchor screens in OVCAR8 and A375 cells; we also performed screens with the PARP inhibitor olaparib. Additionally, we screened a knockout single-cell clone of *PARP1* in the near-haploid cell line HAP1 (ref. [44]), as well as an HAP1 parental control line with another PARP inhibitor, talazoparib. Following sequencing, Z-scores were calculated as before (Supplementary Data 1, Supplementary Data 2, Supplementary Table 1, Supplementary Fig. 11).

Examining top scoring genes across all cell lines and perturbation types (Fig. 5a), we observed that *BRCA1* scored as the 18th ranked gene for synthetic lethality with *PARP1* (Z-score −3.0), and *BRCA2* ranked 54th (Z-score −2.4). Conversely, *PARG*, which catabolizes poly(ADP-ribose), scored as a top buffering gene (rank 2, Z-score 2.8), as has been observed previously[45]. To broadly assess these genetic and small-molecule screens, we used several benchmark gene sets (Supplementary Data 5): a curated set of homologous recombination genes provided by the Wood laboratory ($n = 21$)[46]; the Reactome "DNA Repair" gene set ($n = 106$)[47]; known protein–protein interactors with *PARP1* from BioGrid ($n = 289$)[48]; and a high-confidence set of hits identified by Zimmermann et al.[49] via CRISPR screens for olaparib sensitivity ($n = 73$). Interestingly, the only gene shared in common across all four of these sets is *BRCA1* (Fig. 5b). Across all cell lines and perturbation types (Fig. 5c), we saw statistically significant enrichment for each gene set, with Biogrid showing the least (KS statistic 0.21, one-sided $p$ value $9.6 \times 10^{-12}$) and the Wood curation showing the most (KS 0.80, one-sided $p$ value $1.1 \times 10^{-10}$; note that $p$ value calculation is set-size dependent, and the KS statistic is more appropriate for comparing across gene sets of different sizes).

Comparing genetic knockout of *PARP1* to small-molecule inhibition, we generally observed concordance (Pearson $R = 0.52$) but there were outliers (Fig. 6a). For example, *XRCC1*, which is known to interact directly with *PARP1* (ref. [50]), scored as a top sensitizer for small-molecule PARP inhibition (ranked 4, Z-score −5.6), but is a top buffering gene upon genetic knockout (rank 2, Z-score 2.9). Additionally, *POLB*, which interacts directly with *XRCC1* (ref. [51]), and *HPF1*, which regulates the activity of PARP1 at serine residues[52,53], score as top sensitizers for PARP inhibitors (rank 2, Z-score −6.2 and rank 6, Z-score −5.2 respectively) but do not show any interaction in the genetic knockouts (Z-scores 0.1 and −0.3, respectively). These differences may reflect the distinction between *PARP1* being present in the cell but inhibited by a small molecule, compared to the complete loss of PARP1 protein. Additionally, PARP2 may compensate for PARP1 in the case of genetic knockout, but is likely also inhibited by the small molecules. These results emphasize that although in many cases small-molecule inhibition phenocopies genetic knockout, as with *BCL2L1* and *MCL1* presented above, exceptions can arise.

We also compared the two small molecules used in this study, olaparib and talazoparib, which were included as independent arms in the screens conducted in A375 cells with the Gattinara library, described above (Supplementary Data 3). Overall, the two molecules gave similar results (Pearson $R = 0.50$), with overlapping top hits (Fig. 6b). For example, *ATM*, *RAD54L*, and *NBN* rank in the top 10 of sensitizers with both small molecules,

whereas *TP53*, *PARG*, and *TP53BP1* score as resistance genes (ranks 1, 2, and 20, respectively, averaging across the two inhibitors). Some differences emerge between the small molecules, however. For example, *POLB* is a strong sensitizer with talazoparib (rank 2, Z-score −10.2) but is weaker with olaparib (rank 352, Z-score −2.4). Conversely, loss of *HPF1* strongly synergizes with olaparib (rank 4, Z-score −7.4) and has a weaker phenotype with talazoparib (rank 473, Z-score −2.1). These differences in activity may arise due to differences in PARP-trapping[54].

Interestingly, *PARP1* itself scores as a strong resistance gene with talazoparib (rank 1, Z-score 8.8), but not with olaparib, instead showing a weak sensitization phenotype (rank 152, Z-score −3.0). That *PARP1* loss confers resistance to talazoparib has been reported previously[55], but prior results with olaparib are more complex. *PARP1* loss was shown to confer resistance to olaparib in DT40 and DU145 cells[56], as well as a mouse ES cell clone[55], but *PARP1* knockout did not confer resistance to olaparib in the three cell lines screened by Zimmermann et al.[49], HeLa, RPE1, and SUM149PT. To investigate this further, we first compared the HAP1 *PARP1* knockout clone to parental cells across a range of doses of PARP inhibitors, and we observed resistance (Supplementary Fig. 12). We next examined A375 cells with a competition assay. We transduced A375 populations stably expressing Cas9 with three unique constructs containing both EGFP and an sgRNA targeting *PARP1* at a low MOI, such that 20% of the cells received the construct. We then treated these populations with varying doses of the PARP inhibitors. In this internally controlled assay, EGFP labels *PARP1* knockout cells, and we measured the fraction of EGFP-positive cells over time by flow cytometry (Supplementary Fig. 13a). With all three sgRNAs, we observed that the fraction of *PARP1* knockout cells increased upon treatment with talazoparib, whereas we did not observe an increase in the fraction of EGFP-positive cells treated with olaparib (Supplementary Fig. 13b). Thus, while knockout of *PARG* and *TP53* provided resistance to both talazoparib and olaparib in A375 cells (Fig. 6b), knockout of *PARP1* itself only provided resistance to talazoparib. Further work will be necessary to understand the mechanistic basis of this difference.

Returning to the Brunello screening data, we also observed cell-line-specific differences. For example, the inosine triphosphatase *ITPA* is the top-ranked sensitizer in HAP1 cells (Z-score −9.6) but does not score in A375 (Z-score 0.1) or OVCAR8 (−0.5); this gene has been implicated in DNA damage previously, as ITPA normally prevents base analogs from contributing to the pool of free nucleotides, whereas repair following their incorporation leads to DNA single-strand breaks[57]. Indeed, RNAse H2 enzymes, which act to remove ribonucleotides incorporated into DNA, also score in our screens across all cell lines, consistent with prior screens[49]. Conversely, the DNA ligase *LIG1*, which repairs breaks in DNA during replication, scores strongly in OVCAR8 (rank 4, Z-score −5.1) but not A375 (rank 1420, Z-score −1.1) or HAP1 (rank 17,220, Z-score 1.24). Also in OVCAR8, loss of mitochondrial complex I NDUF genes caused sensitivity to olaparib; 24 of the 80 genes in the GO gene set "oxidative phosphorylation" score in the top 100 (Z-scores < −5.2), whereas this gene set does not show evidence of activity in any other condition (Fig. 6c). Thus, both cell context and mode of inhibition may lead to divergent phenotypes that will require additional investigation to understand mechanistic underpinnings.

Of the top 40 synthetic lethal genes, 16 did not appear in any of the four reference gene sets (Fig. 5b, Supplementary Data 5). Some of these may be false positives, or might be captured by other sources of gene sets with plausible relationships to PARP biology. For example, *SWSAP1* and *ZSWIM7* (rank 6 and 23 overall, respectively) form a complex and are lesser-studied *Rad51*

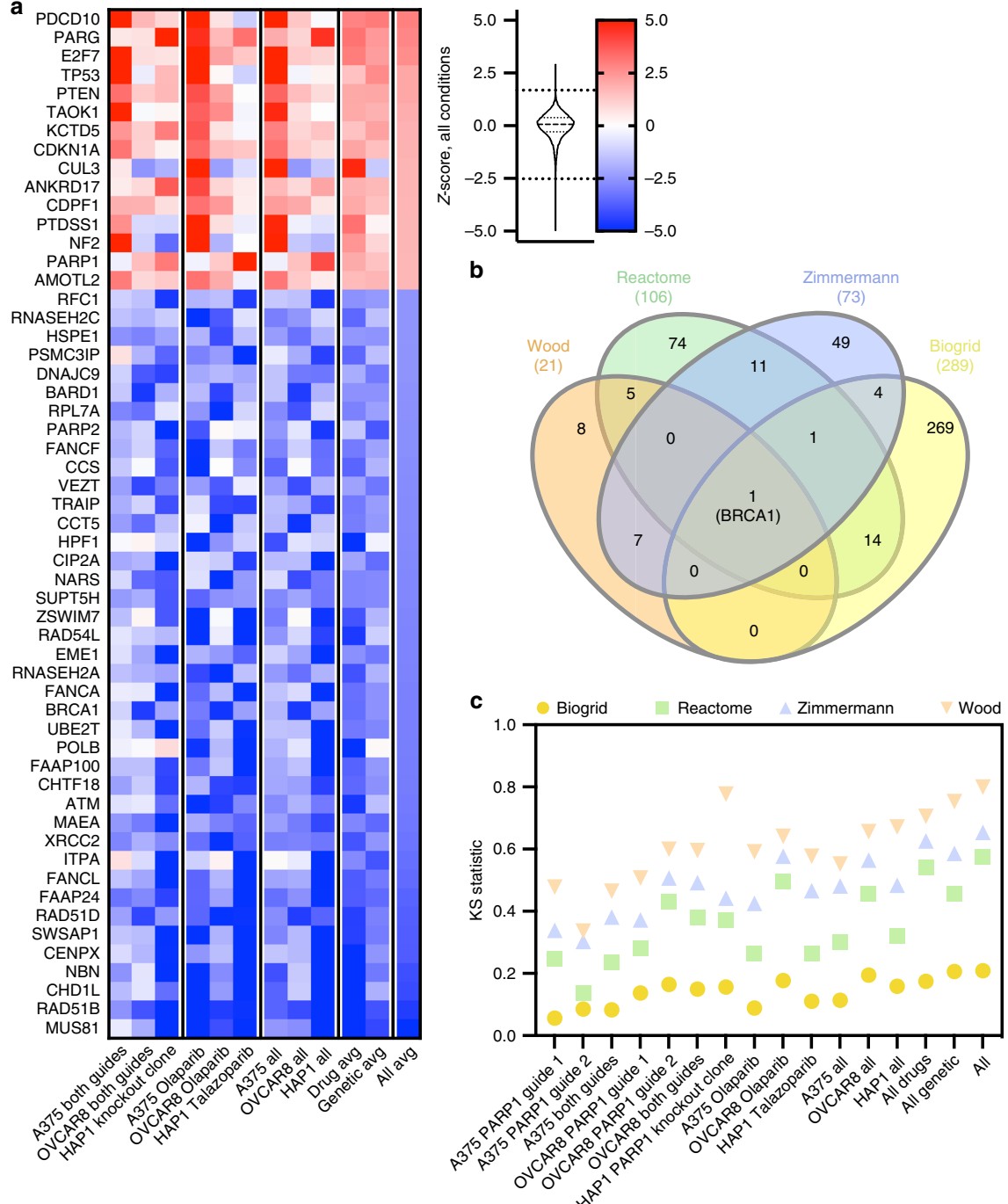

**Fig. 5 Top scoring genes from PARP screens are enriched in related gene sets. a** Top hits by average $Z$-score across all PARP screens. Color scale of $Z$-scores is shown to the right. A violin plot representing the distribution of $Z$-scores is adjacent to the color scale, along with two dotted lines representing the cutoffs for gene hits shown. The 25th, 50th, and 75th percentiles of $Z$-scores are represented by the dotted and dashed lines within the violin plot. The color scale is floored at −5 and ceilinged at 5. **b** Venn diagram of curated gene sets included in the analysis. **c** One-sided KS statistic for each gene set for each screen and averages of various conditions. The value shown represents the alternative hypothesis that the cumulative distribution of genes in the gene set is greater than the distribution of genes not in the set.

paralogues, which appear to be required for efficient homologous recombination, and have been implicated in meiotic recombination[58–60]. Likewise, the genes *CHTF18* and *RFC1* (ranks 14 and 40, respectively) are each core members of distinct replication fork complexes that load the DNA polymerase processivity factor, *PCNA*[61]. Another top hit, *MAEA* (rank 12), is highly correlated in co-essentiality in the DepMap with *UBE2H* ($R = 0.77$), *WDR26* ($R = 0.71$), *YPEL5* ($R = 0.70$), and *GID8* ($R = 0.58$) which rank

558, 86, 131, and 212 in the PARP screens, providing confidence that this novel hit generalizes across cell lines. These five proteins are part of the CTLH E3 ligase complex[62], another member of which, *RANBP9*, has been implicated in sensitivity to DNA damage[63].

Finally, we organized the detected genetic interactions by constructing DepMap and STRING networks (Supplementary Figs. 14 and 15). In both cases, there was more connectivity than

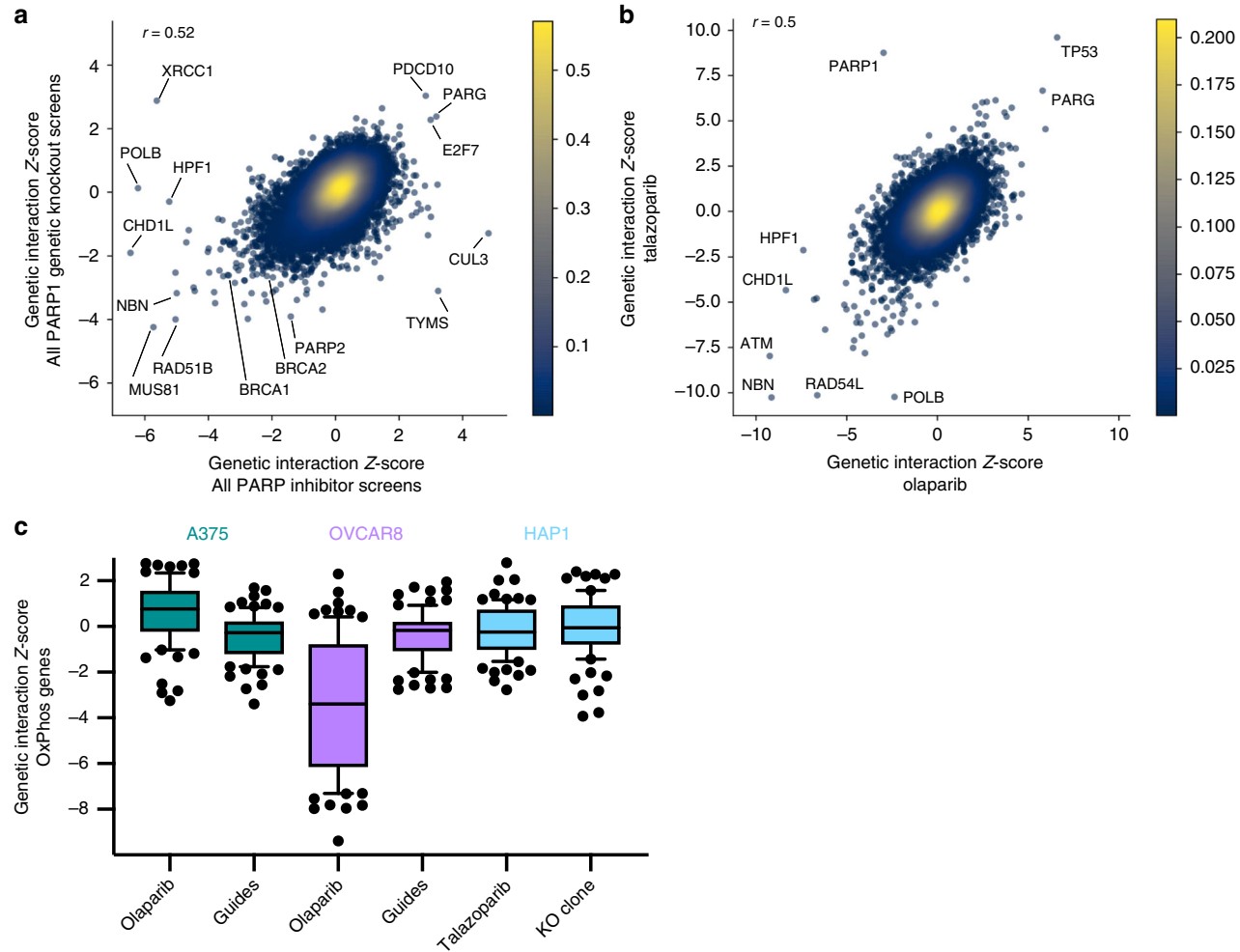

**Fig. 6 Hits from PARP screens agree across contexts with some notable exceptions. a** Average $Z$-scores for PARP inhibitors and genetic knockout perturbations, screened with the Brunello library. Points are colored by density. Pearson correlation coefficient is indicated. **b** $Z$-scores for olaparib and talazoparib perturbations, screened with the Gattinara library in A375 cells. Points are colored by density. Pearson correlation coefficient is indicated. **c** Box plots of $Z$-scores for genes in the GO gene set "oxidative phosphorylation." The box represents the 25th, 50th, and 75th percentiles; whiskers show 10th and 90th percentiles.

expected by chance (Supplementary Fig. 16) with substantial interconnectivity of top hits in both networks (Fig. 7a). Both the DepMap (Fig. 7b) and STRING (Fig. 7c) networks have a cluster containing many Fanconi anemia genes that score as hits in these screens, including *UBE2T*, which was validated in patients as a causal gene for Fanconi anemia[64], as well as *TRAIP*, which has recently been mechanistically connected to the pathway[65]. The resulting networks illustrate how such approaches can suggest functions of less characterized genes.

## Discussion
Here we present a facile approach for generating genome-wide genetic interaction maps for individual genes in cell types of interest using CRISPR technology. Timing the delivery of the anchor perturbation eliminates the need for single-cell cloning, which is typically a major bottleneck for experiments with isogenic cell lines. Importantly, this approach does not generate true isogenic pairs, as DNA double-strand breaks result in a spectrum of indels[66]. Yet, bystander mutations private to single-cell clones are numerous and thus pairs generated by traditional approaches are also not truly isogenic[67]. Indeed, comparisons among *MCL1* knockout clones revealed clone-to-clone heterogeneity.

The success of anchor screens rests heavily on the anchor guide, and thus its performance should be validated before beginning such a screening campaign. Further, an effective screening strategy will be to perform fewer replicates with any one guide, instead employing additional anchor guides to mitigate potential off-target effects of a particular sequence. Although our initial screens used Cas9 from both *S. aureus* and *S. pyogenes*, we subsequently showed that reliance on a single Cas9 can be effective when the perturbation types are the same, such as dual-knockout. The use of two different Cas9s will still be needed when orthogonal activities are desired, such as over-expression of one gene screened against a knockout library.

That co-essentiality data from large-scale genetic screening projects[33,34,38] can be used to generate genetic interaction maps represents a powerful resource for the scientific community[35–37]. However, these large-scale screening projects are the result of many dollars and years of effort, and it is not trivial for individual researchers with, for example, a patient-derived cell line to feed into these pipelines. Further, despite the impressive size of these resources, many tumor types and specific genetic lesions are still poorly represented. Thus, the two-pronged approach described here—perform an anchor screen, then cluster the hits using co-essentiality data—enables researchers to uncover genetic interactions with a gene of interest in a biologically relevant cell

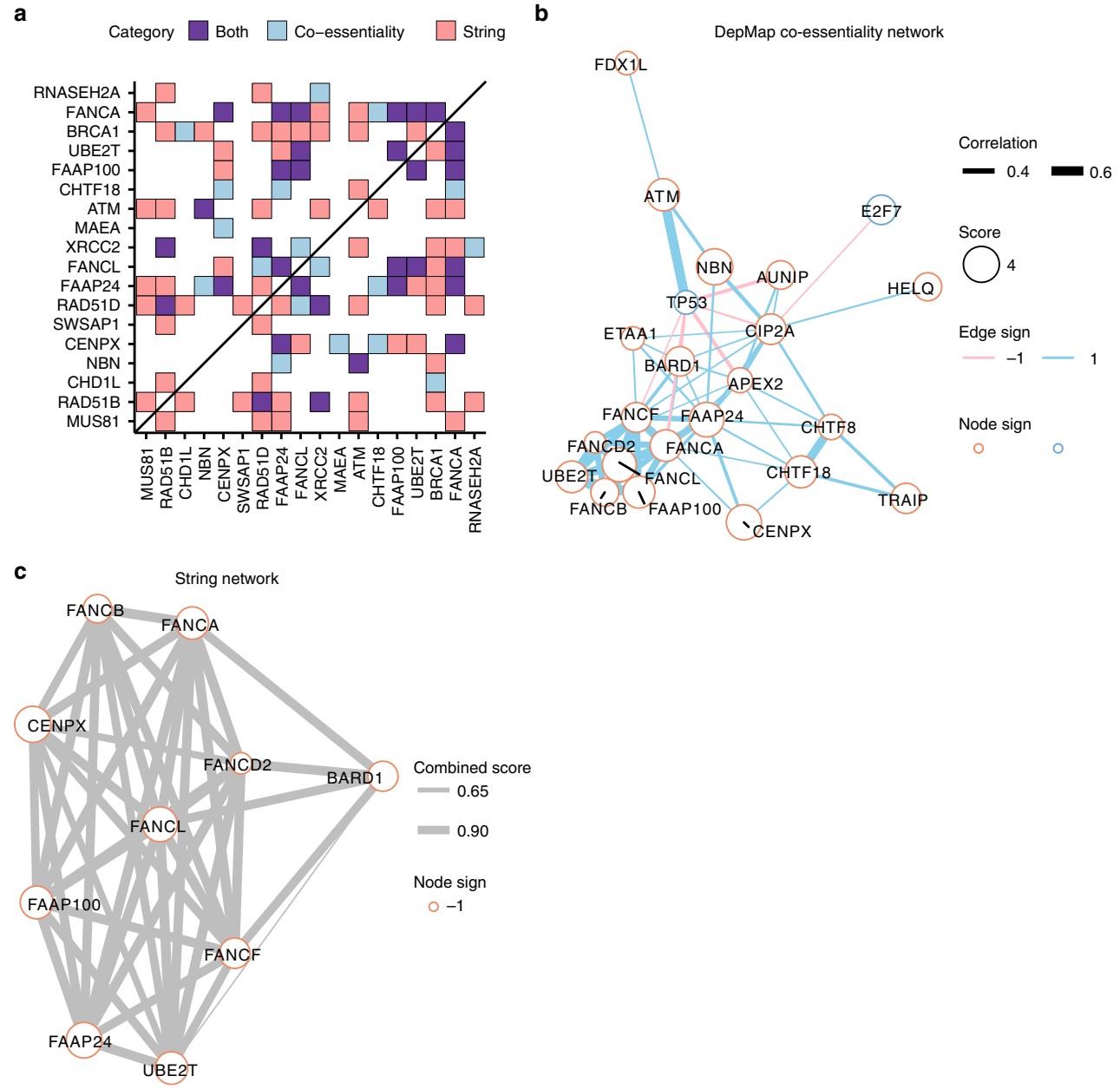

**Fig. 7 PARP screens reveal functionally coherent clusters of genes. a** Interactions between the top 20 gene hits, ranked by absolute Z-score, from DepMap co-essentiality and STRING network sources. Edges are drawn between genes with an absolute correlation greater than 0.2 or a STRING combined score greater than 0.4. Genes are ordered by absolute average Z-score. **b** Cluster with Fanconi anemia genes from the DepMap co-essentiality network. Nodes represent genes and the size of each node is proportional to its average Z-score across all screens. Genes with an absolute average Z-score greater than 2 across PARP1 conditions are included in the network. Edges represent Pearson correlations across co-essentiality profiles in DepMap. Clustering was done by modularity optimization and a single cluster was chosen for visualization. **c** Cluster with Fanconi anemia genes in the STRING network. Nodes are the same as **b**. Edges represent combined score in STRING. Edges are drawn between genes with a STRING combined score greater than 0.4. Clustering was done the same as **a**.

type, but still leverage the data from these large-scale maps to interpret and prioritize the resulting hit genes.

We expect that the anchor screening approach described here will be useful to understand the genetic landscape of a target before a lead candidate small molecule is identified, and to understand the differences between small molecule targeting of a gene and genetic loss-of-function alleles. Related, these screens can identify biomarkers indicating sensitivity to a small molecule. For example, PARP inhibitors have shown clinical efficacy in tumors with *BRCA1* or *BRCA2* mutations[43], which was reca-pitulated in these screens, and their use has been extended to

patients with *ATM* mutations[68], a gene that also scored here. Tumors with mutations in other genes that scored in these screens may also be particularly sensitive to PARP inhibitors. Finally, comparisons between small-molecule inhibitors and genetic knockout reveal that while there is often substantial overlap, the two modes of disrupting a gene are not identical, an observation that has important ramifications for potential drug targets identified by genetic screens.

Although we have presented only double-knockout screens here, simply altering one or both of the Cas9 proteins unlocks a variety of screening possibilities, as we and others have previously

demonstrated in "some-by-some" screens[9,25]. Anchor screens may be particularly powerful when paired with base editing technologies[69], as the introduction of defined gene edits via homologous recombination is at least an order of magnitude less efficient than the generation of knockout alleles, and thus the generation of such isogenic cells has commensurately higher costs. Expanding genetic interaction mapping to include various perturbation types and less tractable cell contexts promises to enhance our capacity to uncover gene function.

## Methods

**Vectors.** Individual sgRNA sequences are provided in Supplementary Table 2. The following vectors were used in the study and are available on Addgene:

pXPR_212 (library vector): U6 promoter expresses customizable Spyo-guide; EFS promoter expresses SaurCas9 and 2A site provides puromycin resistance (Addgene 133457).

pXPR_213 (anchor vector): H1 promoter expresses customizable Saur-guide; EF1a promoter expresses SpyoCas9 and 2A site provides blasticidin resistance (Addgene 133456).

pRDA_186 (Spyo-only anchor vector): U6 promoter expresses customizable Spyo-guide; PGK promoter expresses blasticidin resistance and 2A site provides EGFP (Addgene 133458).

lentiCRISPRv2 (pXPR_023): EF1a promoter expresses SpyoCas9 and 2A site provides puromycin resistance; U6 promoter expresses customizable Spyo-guide (Addgene 52961).

pRosetta_v2: PGK promoter expresses hygromycin resistance, T2A site provides blasticidin resistance, P2A site provides puromycin resistance, and F2A site provides EGFP (Addgene 59700).

pLX_311-Cas9: SV40 promoter expresses blasticidin resistance; EF1a promoter expresses SpyoCas9 (Addgene 96924).

pRDA_118 (modified lentiGuide): U6 promoter expresses customizable Spyo-guide; EF1a promoter provides puromycin resistance (Addgene 133459). This vector is a derivative of the lentiGuide vector, with a modification to the tracrRNA to eliminate a run of four thymidines.

pRDA_103: H1 promoter with two Tet operator (TetO) sites expresses customizable Spyo-guide; short EF1a promoter (EFS) expresses SaurCas9, 2A provides TetR, and 2A provides blasticidin resistance (Addgene 133460).

pXPR_124: EF1a promoter expresses SpyoCas9 and P2A provides EGFP (Addgene 133461).

**Libraries.** The pooled genome-wide library targeting human genes with two guides per gene, Gattinara (Addgene 136986), is available in the pRDA_118 vector, as is the mouse version, Gouda (Addgene 136987). The Brunello library in lenti-CRIPSRv2 (Addgene 73179) was used for the all-Spyo approach.

**Library production.** Oligonucleotide pools were synthesized by CustomArray. *Bsm*BI recognition sites were appended to each sgRNA sequence along with the appropriate overhang sequences (bold italic) for cloning into the sgRNA expression plasmids, as well as primer sites to allow differential amplification of subsets from the same synthesis pool. The final oligonucleotide sequence was thus: 5′-[Forward Primer]CGTCTCA**CACC**G[sgRNA, 20 nt]**GTTT**CGAGACG [Reverse Primer].

Primers were used to amplify individual subpools using 25 µL 2× NEBnext PCR master mix (New England Biolabs), 2 µL of oligonucleotide pool (~40 ng), 5 µL of primer mix at a final concentration of 0.5 µM, and 18 µL water. PCR cycling conditions: 30 s at 98 °C, 30 s at 53 °C, 30 s at 72 °C, for 24 cycles. In cases where a library was divided into subsets unique primers could be used for amplification:

Primer set; forward primer, 5′–3′; reverse primer, 5′–3′
1; AGGCACTTGCTCGTACGACG; ATGTGGGCCCGGCACCTTAA
2; GTGTAACCCGTAGGGCACCT; GTCGAGAGCAGTCCTTCGAC
3; CAGCGCCAATGGGCTTTCGA; AGCCGCTTAAGAGCCTGTCG
4; CTACAGGTACCGGTCCTGAG; GTACCTAGCGTGACGATCCG
5; CATGTTGCCCTGAGGCACAG; CCGTTAGGTCCCGAAAGGCT
6; GGTCGTCGCATCACAATGCG; TCTCGAGCGCCAATGTGACG.

The resulting amplicons were PCR-purified (Qiagen) and cloned into the library vector via Golden Gate cloning with Esp3I (Fisher Scientific) and T7 ligase (Epizyme); the library vector was pre-digested with *Bsm*BI (New England Biolabs). The ligation product was isopropanol precipitated and electroporated into Stbl4 electrocompetent cells (Life Technologies) and grown at 30 °C for 16 h on agar with 100 µg mL$^{-1}$ carbenicillin. Colonies were scraped and plasmid DNA (pDNA) was prepared (HiSpeed Plasmid Maxi, Qiagen). To confirm library representation and distribution, the pDNA was sequenced.

**Lentivirus production.** For small-scale virus production, the following procedure was used: 24 h before transfection, HEK293T cells were seeded in six-well dishes at

a density of 1.5e6 cells per well in 2 mL of DMEM +10% FBS. Transfection was performed using TransIT-LT1 (Mirus) transfection reagent according to the manufacturer's protocol. Briefly, one solution of Opti-MEM (Corning, 66.25 µL) and LT1 (8.75 µL) was combined with a DNA mixture of the packaging plasmid pCMV_VSVG (Addgene 8454, 250 ng), psPAX2 (Addgene 12260, 1250 ng), and the transfer vector (e.g., pLentiGuide, 1250 ng). The solutions were incubated at room temperature for 20–30 min, during which time media was changed on the HEK293T cells. After this incubation, the transfection mixture was added dropwise to the surface of the HEK293T cells, and the plates were centrifuged at 1000 g for 30 min at room temperature. Following centrifugation, plates were transferred to a 37 °C incubator for 6–8 h, after which the media was removed and replaced with DMEM +10% FBS media supplemented with 1% BSA.

A larger-scale procedure was used for pooled library production. Twenty-four hours before transfection, 1.8e7 HEK293T cells were seeded in a 175 cm$^2$ tissue culture flask and the transfection was performed using 6 mL of Opti-MEM, 305 µL of LT1, and a DNA mixture of pCMV_VSVG (5 µg), psPAX2 (50 µg), and 40 µg of the transfer vector. Flasks were transferred to a 37 °C incubator for 6–8 h; after this, the media was aspirated and replaced with BSA-supplemented media. Virus was harvested 36 h after this media change.

**Cell culture.** A375, OVCAR8, and Meljuso cells were obtained from the Cancer Cell Line Encyclopedia several years ago and maintained in-house. HEK293Ts were obtained over a decade ago from ATCC (CRL-3216) and maintained in-house.

HAP1 *PARP1* knockout single-cell clone (HZGHC003943c006) and the unmodified parental line (item C631) were obtained from Horizon Discovery

A375 parental cells from Horizon Discovery, used for comparison to *MCL1* single-cell knockout clones, are catalog number HD PAR-096 (HD clone number 361). *MCL1* knockout clone 1F6 was characterized with a genotype of −/−/−/− and is catalog number HD 118-006 (HD clone number 30928). *MCL1* knockout clone 1B9 was characterized with a genotype of −/−/− and is catalog number HD 118-005 (HD clone number 30972). We confirmed with the vendor that clone 1B9 would have been characterized as +/−/−/− if a fourth, wild-type allele were detected.

All cell lines were routinely tested for mycoplasma contamination and were maintained without antibiotics except during screens, when the media was supplemented with 1% penicillin/streptomycin. Cell lines were kept in a 37 °C humidity-controlled incubator with 5.0% CO2 and were maintained in exponential phase growth by passaging every 2–3 days.

Cells were regularly maintained in antibiotic-free media, except during screens, when cells were maintained in media containing 1% penicillin/streptomycin. The following media conditions and doses of polybrene, puromycin, and blasticidin, respectively, were used:

1. A375: RPMI + 10% FBS; 1 µg mL$^{-1}$; 1 µg mL$^{-1}$; 5 µg mL$^{-1}$
2. HAP1: IMDM + 10% FBS; 4 µg mL$^{-1}$; 1 µg mL$^{-1}$; 5 µg mL$^{-1}$
3. HEK293T: DMEM + 10% FBS; N/A; N/A; N/A
4. Meljuso: RPMI + 10% FBS; 4 µg mL$^{-1}$; 1 µg mL$^{-1}$; 4 µg mL$^{-1}$
5. OVCAR8: RPMI + 10% FBS; 4 µg mL$^{-1}$; 1 µg mL$^{-1}$; 8 µg mL$^{-1}$

Olaparib (10621) was obtained from Cayman Chemical Co. and screened at a dose of 250 nM (in A375) and 500 nM (in OVCAR8). Talazoparib (BMN-673), navitoclax (ABT-263), venetoclax (ABT-199), niraparib (MK-4827), and veliparib (ABT-888) were obtained from Selleckchem. Talazoparib was screened at doses of 7.81 nM (in A375) and 1.95 nM (in HAP1). Navitoclax and venetoclax were both used at a dose of 250 nM. S63845 was a gift from Guo Wei and was screened at 250 nM. A-1331852 (A-6048) was obtained from Active Biochem and was screened at a dose of 250 nM.

**Determination of antibiotic dose.** In order to determine an appropriate antibiotic dose for each cell line, cells were transduced with the pRosetta_v2 lentivirus such that approximately 30% of cells received the construct, and were therefore EGFP+. At least 1 day post-transduction, cells were seeded into six-well dishes at a range of antibiotic doses (e.g. from 0 to 8 µg mL$^{-1}$ of puromycin). The rate of antibiotic selection at each dose was then monitored by performing flow cytometry for EGFP+ cells. For each cell line, the antibiotic dose was chosen to be the lowest dose that led to at least 95% EGFP+ cells after antibiotic treatment for 7 days (for puromycin) or 14 days (for blasticidin and hygromycin).

**Determination of lentiviral titer.** To determine lentiviral titer for transductions, cell lines were transduced in 12-well plates with a range of virus volumes (e.g. 0, 150, 300, 500, and 800 µL virus) with 3e6 cells per well in the presence of poly-brene. The plates were centrifuged at 640g for 2 h and were then transferred to a 37 °C incubator for 4–6 h. Each well was then trypsinized, and an equal number of cells seeded into each of two wells of a six-well dish. Two days post-transduction, puromycin was added to one well. After 5 days, both wells were counted for viability. A viral dose resulting in 30–50% transduction efficiency, corresponding to an MOI of ~0.35–0.70, was used for subsequent library screening.

**Screens.** Transductions were performed with enough cells to achieve a representation of at least 500 cells per sgRNA per replicate, taking into account a

30–50% transduction efficiency. Throughout the screen, cells were split at a density to maintain a representation of at least 500 cells per sgRNA, and cell counts were taken at each passage to monitor growth. Puromycin selection was added 2 days post-transduction and was maintained for 5–7 days. After puromycin selection was complete, each replicate was divided into untreated and, if applicable, small-molecule treatment arms, each at a representation of at least 500 cells per sgRNA. Small-molecule doses used for each cell line are described above. Three weeks after library transduction, cells were pelleted by centrifugation, resuspended in phosphate-buffered saline, and frozen promptly for genomic DNA isolation.

Brunello anchor screens that utilized a Saur-guide as an anchor were established by transducing cells with the pXPR_213 anchor lentiviral vectors, which express a customizable Saur-guide off of the H1 promoter, SpyoCas9 off of the EF1α promoter, and blasticidin resistance from a 2A site. Prior to screening-scale transduction, pXPR_213-expressing cell lines were selected with blasticidin for 14 days. Cell lines expressing pXPR_213 were then transduced with the Brunello library in pXPR_212 in two biological replicates at a low MOI (~0.5).

Secondary library anchor screens that utilized an Spyo-guide as an anchor were established by transducing A375 cells with the pXPR_186 anchor lentiviral vectors, which express a customizable Spyo-guide off of the U6 promoter, blasticidin resistance from the PGK promoter, and a 2A site-expressing EGFP. Prior to screening-scale transduction, pXPR_186-expressing cell lines were selected with blasticidin and monitored for EGFP expression for 14 days. Cell lines expressing pXPR_186 were then transduced with the secondary library in pXPR_023 in two biological replicates at a low MOI (~0.5).

Gattinara screens were executed by transducing A375 cells with the lentiviral vector pLX_311-Cas9, which expresses blasticidin resistance from the SV40 promoter and Cas9 from the EF1α promoter. Prior to screening-scale transduction, pLX_311-Cas9 expressing cell lines were selected with blasticidin for 14 days. Cell lines expressing pLX_311-Cas9 were then transduced with Gattinara in pRDA_118 in two biological replicates at a higher-than-typical MOI (~1.0).

*PARP1* single-cell clone screens in HAP1 were transduced at an early passage with the Brunello library in pXPR_023 in two biological replicates at a low MOI (~0.5).

**Genomic DNA isolation and sequencing**. Genomic DNA (gDNA) was isolated using the Machery Nagel NucleoSpin Blood Maxi (2e7-1e8 cells), Midi (5e6-2e7 cells), or Mini (<5e6 cells) kits as per the manufacturer's instructions. The gDNA concentrations were quantitated by Qubit. For PCR amplification, gDNA was divided into 100 μL reactions such that each well had at most 10 μg of gDNA. Per 96-well plate, a master mix consisted of 150 μL ExTaq DNA Polymerase (Takara), 1 mL of 10× ExTaq buffer, 800 μL of dNTP provided with the enzyme, 50 μL of P5 stagger primer mix (stock at 100 μM concentration), and 2 mL water. Each well consisted of 50 μL gDNA plus water, 40 μL PCR master mix, and 10 μL of a uniquely barcoded P7 primer (stock at 5 μM concentration). For the Spyo-only validation screens in A375 cells and the *MCL1* single-cell clones, the master mix was modified as follows: 150 μL Titanium Taq DNA Polymerase (Takara), 1 mL of 10× Titanium Taq buffer, 800 μL of dNTP (Takara, 4030), 50 μL of P5 stagger primer mix (stock at 100 μM concentration), 500 μL of DMSO, and 1500 μL water. We recommend the latter protocol going forward.

PCR cycling conditions: an initial 1 min at 95 °C; followed by 30 s at 94 °C, 30 s at 52.5 °C, 30 s at 72 °C, for 28 cycles; and a final 10 min extension at 72 °C. P5/P7 primers were synthesized at Integrated DNA Technologies (IDT). PCR products were purified with Agencourt AMPure XP SPRI beads according to the manufacturer's instructions (Beckman Coulter, A63880). Samples were sequenced on a HiSeq2500 HighOutput (Illumina), loaded with a 5% spike-in of PhiX DNA.

Reads were counted by first searching for the CACCG sequence in the primary read file that appears in the vector 5′ to all sgRNA inserts. The next 20 nts are the sgRNA insert, which was then mapped to a reference file of all possible sgRNAs present in the library. The read was then assigned to a condition (e.g. a well on the PCR plate) on the basis of the 8nt barcode included in the P7 primer.

**Screen analysis**. Following deconvolution, the resulting matrix of read counts was first normalized to reads per million within each condition by the following formula: read per sgRNA/total reads per condition × 1e6. Reads per million was then log2-transformed by first adding one to all values, which is necessary in order to take the log of sgRNAs with zero reads. For each sgRNA, the log2-fold-change from plasmid DNA (pDNA) was then calculated.

The log2-fold-changes for each perturbed arm were fit using a natural cubic spline with three degrees of freedom, using the log2-fold-changes of the relevant control arm as reference. We then used the residual from this fit as a phenotypic measure for each guide.

**Synthetic interaction statistical significance**. In order to determine the significance of synthetic interactions at the gene level we averaged the residuals of guides targeting a gene and then calculated a Z-score for these values by subtracting the average residual and dividing by the standard error of all guides. In order to calculate the standard error we took the standard deviation of all guides and divided it by the square root of the number of guides per gene. In doing so, we

assume the distribution of residuals is normal and the average and standard deviation of all guides is representative of the population.

**Network analysis**. All network analyses were done in R. Visualizations were done using the tidygraph and ggraph packages. Network clustering was done using the cluster_louvain function in igraph[70]. We used absolute correlations for co-essentiality and combined scores for STRING as edge weights for the clustering algorithm. Graphs are plotted using the force directed layout in igraph.

**Apoptosis GFP competition assay**. Doxycycline inducible *MCL1*, *BCL2L1*, *MARCH5*, and *WSB2* anchor cell lines were generated by delivering the pRDA_103 vector via a no-spin transduction. Meljuso cells were seeded in a T75 flask in a total volume of 8.6 mL of virus-containing media with polybrene at 0.5 μg mL⁻¹. Flasks were then transferred to an incubator overnight, and the virus-containing media was replaced with fresh media 16–18 h after seeding. Blasticidin selection was added 2 days post-transduction and was maintained for 14 days. Cells were then transduced with pXPR_124 (SpyoCas9-P2A-EGFP) at an MOI of ~0.5, generating a mixed population of EGFP+ and EGFP− cells. After 5 days, cells were treated with 1 μg mL⁻¹ of doxycycline to induce expression of Spyo-guide. On day 7 post infection, cells were treated with 250 nM of either S63845, venetoclax (ABT-263), A-1331852, or navitoclax (ABT-199). The fraction of EGFP-positive cells was monitored for 2 weeks by flow cytometry (BD Accuri C6 Sampler) upon every cell passage. An example gating strategy is provided in Supplementary Fig. 17.

**PARP inhibitor GFP competition assay**. A375 cells were transduced with pLX_311-Cas9 to generate lines stably expressing Cas9. After 2 weeks of selection with blasticidin, vectors delivering EGFP and a guide targeting *PARP1* were introduced, resulting in a heterogeneous population of EGFP-positive and negative cells. Three days post-transduction with guide construct, cells were treated with varying doses of olaparib, talazoparib, niraparib, or veliparib. The fraction of EGFP-positive cells was monitored for 17 days by flow cytometry (BD Accuri C6 Sampler) upon every cell passage.

**Statistical analysis**. All KS tests and Z-scores were calculated in R. Pearson correlation coefficients for density plots were done in Python.

**Reporting summary**. Further information on research design is available in the Nature Research Reporting Summary linked to this article.

## Data availability
The read counts for all screening data and subsequent analyses are provided as Supplementary Data. Fastq files of sequencing are available from the Sequencing Read Archive, accession code SRP217813. Any other relevant data are available from the authors upon reasonable request.

## Code availability
All custom code used for analysis and example notebooks are available on GitHub.

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

## Acknowledgements

We thank Amy Goodale, Hans Reuter, Wei Zhen, Briana Fritchman, and Xiaoping Yang for producing guide libraries and lentivirus; Olivia Bare and Yenarae Lee for logistics support; Matthew Greene, Adam Brown, Doug Alan, Mark Tomko, and Tom Green for software engineering support; the Broad Institute Genomics Platform Walk-up Sequencing group for Illumina sequencing; and the Functional Genomics Consortium for funding support. We thank Sven Rottenberg (Netherlands Cancer Institute, University of Bern) and Steven Jackson (University of Cambridge) for helpful conversations regarding PARP inhibitors.

## Author contributions

Conceived of the study: J.G.D. Executed genetic screens: K.R.S., R.E.H., A.K.S., and C.S. Performed analyses: J.G.D., P.C.D., M.H., and N.S.P. Created visualizations: P.C.D., K.R.S., and A.K.S. Curated data: P.C.D. and M.H. Wrote the manuscript: J.G.D., P.C.D., R.E.H., and A.K.S. Supervised the project: J.G.D.

## Competing interests

J.G.D. consults for Tango Therapeutics, Foghorn Therapeutics, Maze Therapeutics, and Pfizer. All other authors declare no competing interests.
