## [Peer Review File · Nature Communications]

Reviewers' Comments:

Reviewer #1:

Remarks to the Author:

In this study the authors describe a smart way to perform CRISPR screens in paired mutant cell lines without the need for single cell cloning of knock out cells. Screens in isogenic cell lines are very powerful to study gene function because isogenic cells are identical except for one single gene, which allows to immediately draw convincing conclusions from the differences in phenotypic readouts between these isogenic pairs. However, generating isogenic pairs is labor intensive and time consuming.

First, by combining two different endonucleases, SpyoCas9 and SaurCas9, and two subsequent steps of viral transduction, the authors demonstrate this approach is able to discover buffering as well as synthetic lethal genetic interactions. This is shown by screens for the well-validated genes MCL1 and BCL2L1 as anchor genes, and by pharmacological inhibition of each of these genes in both a melanoma and an ovarian cancer cell line. In this approach first a vector that delivers the SpyoCas9 gene and an SaurCas9 guide to the anchor gene is transduced after which a vector which delivers the SaurCas9 gene and the library of SpyoCas9 guides is transduced. This allows for simultaneous knockout of the anchor gene and the gene targeted by the library. These screens were able to identify expected synthetic lethality or resistance. But also new genes synthetically lethal with BCL2L1 and MCL1 are discovered but without further biochemical characterization. Subsequent network analyses were performed, which revealed a high level of connectivity between the top identified genes.

Although several vector designs using different endonucleases for creating double knockout have been described in the literature, this study next describes the use of only SpyoCas9 to generate both knockouts, which makes it easier to use (in terms of cloning) than the yet described methods. Double knock-out is achieved by first transducing the Spyo anchor guide and then the library containing both the SpyoCas9 gene and the Spyo library guides. In this study this screen was applied as a secondary screen for MCL1 using less genes with more guides and more anchor guides. These secondary screens validated hits from the primary screen, but also identified more genes such as three members from the cullin-RING ubiquitin ligase complex or HSP90AB1.

Last, an anchor screen with PARP1 and PARP inhibitors was performed next to a screen in a knock out single gene clone of PARP1 in HAP1 cells. Synthetic lethality between PARP1 and BRCA1 is well-established, but BRCA1 scored only as the 18th ranked gene in the anchor screen, while MUS81 and RAD51B ranked as top resistance genes. Although there was general concordance between knockout and small molecule inhibition screens, there were some exceptions. Comparison of two PARP1 inhibitors olaparib and talazoparib gave overall similar results between both compounds, although there were also exceptions. A remarkable difference however, was that PARP1 scored as a strong resistance gene with talazoparib but shows sensitization with olaparib, which are both PARP1 inhibitors. There were also cell-line specific differences between the anchor screens.

General comments:

This is an interesting study and a very well written manuscript. The authors describe a very simple but smart way of overcoming single cell cloning for screening genetic networks using isogenic cell lines in a "one-by-all" screening format.

In the PARP1 anchor screen, authors have used a single cell PARP1 knockout clone of HAP1 cells. Why was this cell line chosen? It would have been better if a single cell clone of either A375 or OVCAR8 cells was used instead of HAP1 because the anchor screen was done in A375 and OVCAR8 cells. Especially because authors observe cell-line specific differences between identical screens in different cell lines. Using the same cell line would allow for a correct comparison between the single cell clone isogenic screen with the in this study proposed anchor screen. Especially because this is one of the main messages of the study. It would have been nice if a single cell clone

knockout screen was compared to the anchor screen in the same cell line.

Two small molecule PARP1 inhibitors were compared in the screens and differences were observed. However, I find the explanation for these differences given by the authors in lines 321, 322 that talazoparib is a stronger PARP1 inhibitor the least plausible. First, it is impossible to draw conclusions on differences between these compounds based on the performed screens because they were not done in the same cell line. Olaparib was tested in A375 and OVCAR8 cells while talazoparib was tested in HAP1 cells. From Fig 5a it is clear that the results for olaparib treatment already differs between A375 cells and OVCAR8 cells (see CUL3, NF2, ...). Also, authors mention that they observed cell-line specific differences in their screens (lines 333). Therefore, the difference seen between olaparib and talazoparib may be related to the difference in cell line used and not to the difference between the compounds. Especially for PARP1, the target of both compounds, which scores as a strong resistance gene with talazoparib but shows sensitization with olaparib. This difference is too big to be explained by just difference in potency between the two compounds. Especially if both compounds have the same mechanism of action.

Specific comments:

- Line 151-153: strong buffering for TP53 in Meljuso cells but not in OVCAR8 cells (Fig 2c). Are OVCAR8 may be p53 null?
- Line 229-231: Because two separate vectors are used but the same promoter this system has less competition than approaches that use different promoters? The same promoter competes for the same transcription factors while different promoters don't, so I would expect less competition when using different promoters. Is there less competition when one uses separate vectors instead of one?
- Line 219: It is the first time/study the Gattinara library is mentioned/used. Can the authors give more information on this library, such as guide design? Especially because the guides are different from the ones used in the Brunello library and because the Brunello has become a standard library.
- Line 233: What is the rationale for taking 390 hit genes and 857 non-scoring genes? What was taken as cut-off?
- Line 295: XRCC1 is a top sensitizer for pharmacological PARP inhibition. It not shown in Fig 5a?
- Line 333: may be ITPA is not expressed in A375 or OVCAR8 cells?

Reviewer #2:

Remarks to the Author:

Genetic screens in isogenic mammalian cell lines without single cell cloning.

DeWeirdt et al

Summary of work and major conclusions

The authors describe a new system for conducting genetic interaction screens in cell lines. They provide several examples of screens using this strategy and compare them to single knockout screens using drugs targeting the "anchor" gene products.

Is the question addressed important?

Genetic interaction screens are a crucial technology for understanding how cells work and for discovery of new targets in genetic diseases such as cancer. Carrying out such screens is time-consuming, so improvements to the technology will widen its application.

Are the conclusions novel and will they influence thinking in the field?

The screen datasets presented will be useful to others in the field considering similar screens. There are few new biological insights from the screens performed, and these are not followed up

except by integrating data from further screens. This is fundamentally a methods paper but will be a valuable addition to the literature.

Quality of the data provided

Data quality is generally good and the Supplementary Data are clear and complete.

Major points

A number of different screening methods and libraries are used (Spyo + Saur anchor, Saur only anchor, Spyo only anchor, Brunello/Gattinara/subgenomic libraries). This gives the paper a disjointed feel. Including a single discussion subsection with the pros and cons of each approach (currently scattered throughout the results section) would help to integrate the diverse approaches used. The authors appear to be favouring the Spyo query - Saur anchor approach and describe several advantages to this (principally speed). However, these rely on a well-characterised Saur guide being available - available in this case, but the characterisation of which for a new gene would add significant time. There is also no direct comparison of the anchor method to an isogenic screen - the issue the authors were addressing - so it isn't completely clear that this method is an improvement, except in terms of time. This could be accomplished by carrying out a screen using the HAP1 PARP1 line, for example.

Where possible, adding some functional QC data (e.g. dropout of known essential genes) would help. As far as I know the Gattinara library has not been previously described so some data to allow the reader to assess its performance compared to Brunello independently of the screen results should be included.

The authors discuss the drawbacks of relying on a single guide in the discussion, and also raise good points about the true isogenicity of knockout clones. However, the authors have the data to address this in a more direct way, as two guides were used for the PARP1 anchor screens in A375 and OVCAR8 cells. Some analysis of how reproducible these screens were would help to address this point of whether a single anchor guide (as in the BCL2L1 and MCL1 screens) might be sufficient. At first glance they seem reasonably similar for OVCAR8 but not so much for A375, but in each case combining the results for both guides yields a list that is enriched in known PARP dependencies. It seems that these data suggest that screening using multiple anchor guides would be important to minimise false positives but I think the paper would benefit from a proper analysis of this (e.g. cross correlations and comparison of hit lists for each guide). The heatmap Figures (2c, d, 5a etc) only show the genes that score reproducibly across all conditions and so do not really address the question of how the results would look with just a single guide.

There is no follow-up biological analysis for any of the potentially novel findings, meaning that conclusions regarding their importance are extremely speculative. In particular, the suggestion that PARP1 disruption causes talazoparib resistance but olaparib sensitivity is not consistent with published data in which PARP1 mutant cells are similarly resistant to talazoparib and olaparib in a variety of contexts (e.g. Murai et al 2014, Pettitt et al 2018). Some differences between PARP1 in olaparib and talazoparib CRISPR screens have been observed in previous studies but are likely consequences of screen design (which may relate to trapping/relative toxicity as suggested) rather than biologically significant. One important detail missing from the methods is what concentration (in absolute and surviving fraction terms) of drug was used in the different screens.

Minor points

Figure 1b - are any of the screens actually as shown here - a genetic interaction screen followed by a small molecule treatment?

Figure 2a - plot the fitted model used to calculate the residuals.

Figure 2b - how were the genes shown here selected?

Figure 3c would be clearer if the diagonal symmetry was emphasised - by a diagonal line or removing one diagonal half of the figure.

Line 349 - C20orf196 (not 192)

Reviewers' comments:

Reviewer #1 (Remarks to the Author):

In this study the authors describe a smart way to perform CRISPR screens in paired mutant cell lines without the need for single cell cloning of knock out cells. Screens in isogenic cell lines are very powerful to study gene function because isogenic cells are identical except for one single gene, which allows to immediately draw convincing conclusions from the differences in phenotypic readouts between these isogenic pairs. However, generating isogenic pairs is labor intensive and time consuming.

First, by combining two different endonucleases, SpyoCas9 and SaurCas9, and two subsequent steps of viral transduction, the authors demonstrate this approach is able to discover buffering as well as synthetic lethal genetic interactions. This is shown by screens for the well-validated genes MCL1 and BCL2L1 as anchor genes, and by pharmacological inhibition of each of these genes in both a melanoma and an ovarian cancer cell line. In this approach first a vector that delivers the SpyoCas9 gene and an SaurCas9 guide to the anchor gene is transduced after which a vector which delivers the SaurCas9 gene and the library of SpyoCas9 guides is transduced. This allows for simultaneous knockout of the anchor gene and the gene targeted by the library. These screens were able to identify expected synthetic lethality or resistance. But also new genes synthetically lethal with BCL2L1 and MCL1 are discovered but without further biochemical characterization. Subsequent network analyses were performed, which revealed a high level of connectivity between the top identified genes.

Although several vector designs using different endonucleases for creating double knockout have been described in the literature, this study next describes the use of only SpyoCas9 to generate both knockouts, which makes it easier to use (in terms of cloning) than the yet described methods. Double knock-out is achieved by first transducing the Spyo anchor guide and then the library containing both the SpyoCas9 gene and the Spyo library guides. In this study this screen was applied as a secondary screen for MCL1 using less genes with more guides and more anchor guides. These secondary screens validated hits from the primary screen, but also identified more genes such as three members from the cullin-RING ubiquitin ligase complex or HSP90AB1.

Last, an anchor screen with PARP1 and PARP inhibitors was performed next to a screen in a knock out single gene clone of PARP1 in HAP1 cells. Synthetic lethality between PARP1 and BRCA1 is well-established, but BRCA1 scored only as the 18th ranked gene in the anchor screen, while MUS81 and RAD51B ranked as top resistance genes. Although there was general concordance between knockout and small molecule inhibition screens, there were some exceptions. Comparison of two PARP1 inhibitors olaparib and talazoparib gave overall similar results between both compounds, although there were also exceptions. A remarkable difference however, was that PARP1 scored as a strong resistance gene with talazoparib but shows

sensitization with olaparib, which are both PARP1 inhibitors. There were also cell-line specific differences between the anchor screens.

General comments:

This is an interesting study and a very well written manuscript. The authors describe a very simple but smart way of overcoming single cell cloning for screening genetic networks using isogenic cell lines in a “one-by-all” screening format.

Thank you, we are glad you found the manuscript overall well-presented.

In the PARP1 anchor screen, authors have used a single cell PARP1 knockout clone of HAP1 cells. Why was this cell line chosen? It would have been better if a single cell clone of either A375 or OVCAR8 cells was used instead of HAP1 because the anchor screen was done in A375 and OVCAR8 cells. Especially because authors observe cell-line specific differences between identical screens in different cell lines. Using the same cell line would allow for a correct comparison between the single cell clone isogenic screen with the in this study proposed anchor screen. Especially because this is one of the main messages of the study. It would have been nice if a single cell clone knockout screen was compared to the anchor screen in the same cell line.

To answer the question from the reviewer directly, we originally used the PARP1 knockout clone of HAP1 cells because it was an already-existing reagent that could be purchased from Horizon Discovery, as we did not want to go through the process of single cell cloning ourselves – as the reviewer notes, this is a labor and time intensive process.

We agree with the reviewer’s suggestion for a direct comparison of a single cell clone knockout screen to an anchor screen in the same cell line, and reviewer 2 made a similar comment. Thus, in the revised manuscript we have included screens conducted with two single cell clone knockouts of MCL1, which we commissioned from Horizon Discovery. Please note that, unlike the PARP1 knockout clones that could be purchased off-the-shelf, the generation of MCL1 knockout clones in A375 cells was a custom order that cost \$23,000 and took 5 months to generate. We had hoped to receive them in time for the first submission, but timelines were unclear, and felt the manuscript was ready to share with the scientific community in the absence of these data. We are now pleased to include these comparisons in the revised submission.

Notably, one clone was characterized by the vendor as MCL1 $-/-$, whereas a second was characterized as MCL1 $-/-/-$. The genotyping performed by the vendor did not indicate a remaining wildtype allele in the former clone. We first characterized these clones in our hands by dosing with a BCL2L1 inhibitor. The $-/-/-$ clone showed far more sensitivity than the $-/-$ clone, suggesting that a wildtype allele remained in the $-/-$ line, underscoring the difficulty of properly characterizing single cell clones.

We then screened both of these lines and compared to our previous MCL1 anchor and small molecule inhibitor (S63845) screens. Generally, top hits were consistent across the three approaches used (small molecule inhibition, anchor screen, single cell clone), with BCL2L1 and WSB2 scoring as top synthetic lethal hits. However, differences emerged. For example, comparing the two single cell clones, DUSP4 scored as a synthetic lethal gene in the -/- line but a resistance gene in the -/-/- line. This may be a true biological effect related to the gene dosage of MCL1. However, this may also be an artifact of single cell cloning, in that each clone contains private mutations or epigenetic alterations. Achieving statistical significance to fully distinguish between these two possibilities would require the generation of many more single cell clones, and thus is outside the scope of what is feasible at this time. Nevertheless, this comparison highlights some of the difficulties of interpretation when working with single cell clones. These additions to the manuscript appear in **Fig. 4** of the revision.

Two small molecule PARP1 inhibitors were compared in the screens and differences were observed. However, I find the explanation for these differences given by the authors in lines 321, 322 that talazoparib is a stronger PARP1 inhibitor the least plausible. First, it is impossible to draw conclusions on differences between these compounds based on the performed screens because they were not done in the same cell line. Olaparib was tested in A375 and OVCAR8 cells while talazoparib was tested in HAP1 cells. From Fig 5a it is clear that the results for olaparib treatment already differs between A375 cells and OVCAR8 cells (see CUL3, NF2, ...). Also, authors mention that they observed cell-line specific differences in their screens (lines 333). Therefore, the difference seen between olaparib and talazoparib may be related to the difference in cell line used and not to the difference between the compounds.

It is true that with the initial screens with the Brunello library, presented in **Fig. 5**, we were not able to directly compare talazoparib and olaparib because they were screened in different cell lines. However, with the Gattinara library, we directly compared talazoparib and olaparib in A375 cells (presented in **Fig. 6b**), and it is in this context -- a direct comparison of the two small molecules in the same cell line -- that we suggested observed differences “may be due” to the well-documented differences in PARP-trapping activity of these small molecules.

Especially for PARP1, the target of both compounds, which scores as a strong resistance gene with talazoparib but shows sensitization with olaparib. This difference is too big to be explained by just difference in potency between the two compounds. Especially if both compounds have the same mechanism of action.

It is worth noting that our results are consistent with the Zimmermann genome-wide screen, which treated three different cell lines with olaparib, and did not see loss of PARP1 confer resistance (this screen did not examine talazoparib).

Cell line treated with Olaparib	PARP1 normZ score	PARP1 pval	PARP1 Rank
HeLa	-0.016	0.493	7701
SUM149PT	-1.461	0.072	2481
RPE1-hTERT	1.556	0.94	13570

Data from Zimmermann et al.

Reviewer 2 raised this topic as well, and had additional questions; please see our response below, as it includes new experimental data.

Specific comments:

- Line 151-153: strong buffering for TP53 in Meljuso cells but not in OVCAR8 cells (Fig 2c). Are OVCAR8 may be p53 null?

The Cancer Cell Line Encyclopedia notes a damaging TP53 splice site mutation in OVCAR8 cells but no mutations in Meljuso, so this is a plausible explanation for this difference. Furthermore, data from the Dependency Map (www.depmap.org) show that TP53 knockout causes an increase in proliferation in Meljuso cells (CERES score of 1.05). OVCAR8, however, shows no increase or decrease in proliferation upon TP53 knockout (CERES -0.02), further suggesting they are already loss-of-function for TP53 activity.

- Line 229-231: Because two separate vectors are used but the same promoter this system has less competition than approaches that use different promoters? The same promoter competes for the same transcription factors while different promoters don't, so I would expect less competition when using different promoters. Is there less competition when one uses separate vectors instead of one?

This is certainly possible. Many previous combinatorial approaches locate the two pol III promoters on the same vector, and competition may be enhanced by their close proximity. In contrast, the two pol III promoters used here are located on separate vectors, and thus their integration sites are unlikely to be near each other in the genome. There are already at least 9 copies of the U6 promoter in the human genome, of which at least 5 have been shown to be active (PMID: 12711679), suggesting that transcription factors for these promoters are reasonably abundant.

- Line 219: It is the first time/study the Gattinara library is mentioned/used. Can the authors give more information on this library, such as guide design? Especially because the guides are different from the ones used in the Brunello library and because the Brunello has become a standard library.

Yes, this is a new library, and we are happy to provide more description. We have added **Supplementary Note 1** that focuses on this specifically, so those who are interested in details can read more in one coherent document, without breaking the flow of the main narrative in the manuscript.

- Line 233: What is the rationale for taking 390 hit genes and 857 non-scoring genes? What was taken as cut-off?

In an initial analysis of the screens, hits were selected using a p-value cut-off of 10^{-4} . We subsequently refined our data processing and analysis approach during preparation for submission, and thus the exact p-value cut-off based on the finalized analysis pipeline is slightly different.

- Line 295: XRCC1 is a top sensitizer for pharmacological PARP inhibition. It not shown in Fig 5a?

XRCC1 has some very interesting behavior, as it provides strong resistance to PARP1 knockout HAP1 cells, but strong sensitivity to talazoparib. Since **Fig 5a** is plotted by the average of all conditions, these effects essentially cancel each other out and thus XRCC1 does not appear in that figure. We believe that providing the general answer (i.e. average across all conditions) is the most appropriate display item for a main figure, but we do highlight XRCC1 specifically on other figures. Further, the underlying data are all available as Supplementary Data, so anyone can recreate figures based on alternative selection criteria.

- Line 333: may be ITPA is not expressed in A375 or OVCAR8 cells?

There are no mutations for ITPA indicated in CCLE, and mRNA levels are similar across cell lines (6.83 TPM in A375 and 6.53 TPM in OVCAR8). Thus, there is not a trivial explanation for the difference in phenotype observed across cell lines.

Reviewer #2 (Remarks to the Author):

Genetic screens in isogenic mammalian cell lines without single cell cloning.
DeWeirdt et al
Nature Communications

Summary of work and major conclusions

The authors describe a new system for conducting genetic interaction screens in cell lines. They provide several examples of screens using this strategy and compare them to single knockout screens using drugs targeting the "anchor" gene products.

Is the question addressed important?

Genetic interaction screens are a crucial technology for understanding how cells work and for discovery of new targets in genetic diseases such as cancer. Carrying out such screens is time-consuming, so improvements to the technology will widen its application.

Are the conclusions novel and will they influence thinking in the field?

The screen datasets presented will be useful to others in the field considering similar screens. There are few new biological insights from the screens performed, and these are not followed up except by integrating data from further screens. This is fundamentally a methods paper but will be a valuable addition to the literature.

Quality of the data provided

Data quality is generally good and the Supplementary Data are clear and complete.

Thank you, we do strive to make sure data are usable, especially so that experts from specific biological areas can examine the results closely.

Major points

A number of different screening methods and libraries are used (Spyo + Saur anchor, Saur only anchor, Spyo only anchor, Brunello/Gattinara/subgenomic libraries). This gives the paper a disjointed feel. Including a single discussion subsection with the pros and cons of each approach (currently scattered throughout the results section) would help to integrate the diverse approaches used. The authors appear to be favouring the Spyo query - Saur anchor approach and describe several advantages to this (principally speed). However, these rely on a well-characterised Saur guide being available - available in this case, but the characterisation of which for a new gene would add significant time.

We are happy to bring this together more cohesively, and have updated the text to guide readers through the pros and cons of each approach more explicitly.

There is also no direct comparison of the anchor method to an isogenic screen – the issue the authors were addressing – so it isn't completely clear that this method is an improvement, except in terms of time. This could be accomplished by carrying out a screen using the HAP1 PARP1 line, for example.

Reviewer 1 made a similar point, and we have added a screen in a newly-created MCL1-knockout single cell clone, commissioned from Horizon Discovery for this purpose. Please see our response to R1 for a full overview of these new experimental results.

Where possible, adding some functional QC data (e.g. dropout of known essential genes) would help. As far as I know the Gattinara library has not been previously described so some data to allow the reader to assess its performance compared to Brunello independently of the screen results should be included.

Reviewer 1 made a similar point, and we have added **Supplementary Note 1** with a much fuller description of the Gattinara library and benchmarking of results.

The authors discuss the drawbacks of relying on a single guide in the discussion, and also raise good points about the true isogenicity of knockout clones. However, the authors have the data to address this in a more direct way, as two guides were used for the PARP1 anchor screens in A375 and OVCAR8 cells. Some analysis of how reproducible these screens were would help to address this point of whether a single anchor guide (as in the BCL2L1 and MCL1 screens)

might be sufficient. At first glance they seem reasonably similar for OVCAR8 but not so much for A375, but in each case combining the results for both guides yields a list that is enriched in known PARP dependencies. It seems that these data suggest that screening using multiple anchor guides would be important to minimise false positives but I think the paper would benefit from a proper analysis of this (e.g. cross correlations and comparison of hit lists for each guide). The heatmap Figures (2c, d, 5a etc) only show the genes that score reproducibly across all conditions and so do not really address the question of how the results would look with just a single guide.

We have included scatter plots as **Supplementary Fig. 11** (also included here, to the right) comparing the Z-scores between PARP1 guides in OVCAR8 and A375 to aid readers in understanding the performance of this approach. We see good correspondence between these guides in OVCAR8 (Pearson R = 0.67), and weaker correlations in A375 (Pearson R = 0.39). Note that OVCAR8 was more enriched across all four gold standard lists and perturbation types compared with A375, consistent with these results. The hit lists for each guide is provided in the Supplementary Data.

We have also compared the performance of three MCL1 guides used in the secondary library. Here, correlation at the gene level ranges from a Pearson R of 0.66 - 0.71 in pairwise comparisons, presented in **Supplementary Fig. 10**:

There is no follow-up biological analysis for any of the potentially novel findings, meaning that conclusions regarding their importance are extremely speculative. In particular, the suggestion that PARP1 disruption causes talazoparib resistance but olaparib sensitivity is not consistent with published data in which PARP1 mutant cells are similarly resistant to talazoparib and olaparib in a variety of contexts (e.g. Murai et al 2014, Pettitt et al 2018). Some differences

between PARP1 in olaparib and talazoparib CRISPR screens have been observed in previous studies but are likely consequences of screen design (which may relate to trapping/relative toxicity as suggested) rather than biologically significant. One important detail missing from the methods is what concentration (in absolute and surviving fraction terms) of drug was used in the different screens.

Reviewer 1 raised similar questions. First, it is worth noting that the lack of resistance to olaparib upon PARP1 knockout is consistent with the Zimmermann et al. genome-wide screens, which treated three different cell lines with olaparib (this study did not examine talazoparib). In contrast, we note that Pettitt et al. (Nature Communications, 2018) isolated a mouse ES cell clone that showed resistance to both olaparib and talazoparib. Likewise, Murai et al. (Mol. Cancer Ther., 2014) observed resistance to olaparib, and rucaparib in DT40 and DU145 cells. Thus, it appears that resistance to olaparib may be dependent on cell context.

To gain further insight into these differences, we have added experimental data, and we apologize for omitting these drug dose data in the initial submission. First, using the HAP1 PARP1 knockout single cell clones, we performed an ATP viability assay, compared to unmodified parental cells. Across 4 different PARP inhibitors we saw resistance (note that we do not use niraparib or veliparib in this manuscript, but include the data for the sake of completeness). We also include our dose-response curves in A375 cell for talazoparib and olaparib. These data are now provided as **Supplementary Fig. 12** and are shown to the right for convenience. Note that olaparib was screened in A375 at a dose of 250 nM and talazoparib was screened at doses of 7.81 nM (in A375) and 1.95 nM (in HAP1). All of these doses were selected based on these dose-response curves to achieve only marginal impacts on cell growth (~10% inhibition) in the 3-day time frame of the ATP assay such that we would be powered to see additional drop-out (i.e. sensitization) in the three-week-long genetic screen.

Cell line treated with Olaparib	PARP1 normZ score	PARP1 pval	PARP1 Rank
HeLa	-0.016	0.493	7701
SUM149PT	-1.461	0.072	2481
RPE1-hTERT	1.556	0.94	13570

Data from Zimmermann et al.

Next, we conducted additional experiments in A375 cells. Since we do not have a PARP1 knockout single cell clone, we performed a competition assay, in which all cells carry a Cas9

construct, and then a PARP1 guide which carries an EGFP marker is delivered at low MOI. Thus, EGFP+ cells have PARP1 knockout, whereas dark cells, by virtue of having no guide, are PARP1 wildtype. Enrichment in the fraction of EGFP+ cells over time thus indicates resistance to PARP inhibition.

We performed this experiment with three unique PARP1 guides, and in all cases, observed clear resistance to talazoparib, but no resistance to olaparib, even at 1 μ M, a dose 4x higher than the dose used in pooled screens in A375 cells. Importantly, a dose of 250 nM was sufficient to identify other mechanisms of resistance in A375 cells in the screens, such as loss of TP53 and PARG (Fig. 6). Thus, it cannot be the case that either a) too low a dose of olaparib was used to observe resistance phenotypes, or b) that the PARP1 guides did not work in this screen, as they do provide resistance to talazoparib.

Even though we cannot at this time offer an explanation as to why we observe these differences, we do not believe them to be a technical artifact of the large-scale genetic screens, but rather are true biological differences across cell lines and small molecules.

Minor points

Figure 1b - are any of the screens actually as shown here - a genetic interaction screen followed by a small molecule treatment?

No, we have not actually done that in this study, except in the control case where we wanted to compare genetic knockout to small molecule inhibition, in which case the small molecule treated arm received a control guide. Adding a small molecule in addition to a guide targeting a gene of interest is completely possible, of course.

Figure 2a - plot the fitted model used to calculate the residuals.

We have updated this figure to include the fit line.

Figure 2b - how were the genes shown here selected?

Top hits in each direction (no filtering), but with an arbitrary cut-off for how many to show, dictated by space limitations. All the analyses are available in the Supplementary Data.

Figure 3c would be clearer if the diagonal symmetry was emphasised - by a diagonal line or removing one diagonal half of the figure.

A diagonal line has been included to emphasize symmetry.

Line 349 - C20orf196 (not 192)

This is an impressive catch, thank you!

Reviewers' Comments:

Reviewer #1:

Remarks to the Author:

Authors now have included a direct comparison of the anchor screen to single cell knock-out clones. Authors write there is "general agreement between the approaches" (line 315), but I do not fully agree with this wording as can be seen in Figures 4D,E,F (R 0.34). Also in the lines before this statement authors do acknowledge the differences. As authors also mention, there can be many reasons for these differences. One reason could be that it is the result of long term adaptation of the single cell clones; some genes with a low score in the anchor screens are sensitizing in the "adapted" single cell clones (e.g. PGD), which may be the result of the adaptation. RNASeq comparison or expression level measurements of these genes between a single cell clone and an acute knock-out cell, may help to explain this phenomenon.

The differences between PARP guides in olaparib and talazoparib screens in A375 cells remains puzzling. It appears that next to olaparib, PARP1 guides are also unable to induce resistance to the other PARP inhibitors niraparib and veliparib (Supp Figure 13). This raises the question whether these compounds are really inhibiting A375 cells and if so, do they do this through targeting PARP1 or off target mechanisms? Dose-response curves in these cells do not follow the typical sigmoidal shape, in contrast to the HAP1 cells (Supp Figure 12) or HeLa cells (Zimmerman et al.). Are these A375 cells a good model to study these inhibitors? In their response, authors argue that the dose of 250 nM was sufficient to identify other mechanisms of resistance in A375 cells in the screens, such as loss of TP53 and PARG. This argument is true if they are also able to validate these genes in the same way as they did for the 3 different PARP guides.

minor comment: I believe the correct word to describe the entry of replication deficient lentiviral vectors in cells is "transduction" and not "infection", infection is generally used for fully replicative competent viruses.

check supp figures 2-7 as some annotations disappeared

Please include in the figure legend of Figure 6b the cell line used

Reviewer #2:

Remarks to the Author:

I think the authors have done an excellent job in addressing each of the comments made in the first round of review. The addition of new data is great as are the changes to the text. We'll have to agree to disagree about the PARP1 vs PARP inhibitor issue and think it's important that their data on this is out there !

Reviewer #1 (Remarks to the Author):

Authors now have included a direct comparison of the anchor screen to single cell knock-out clones. Authors write there is "general agreement between the approaches" (line 315), but I do not fully agree with this wording as can be seen in Figures 4D,E,F (R 0.34).

We are happy to rephrase this to remove the vague phrase "general agreement." We have rephrased to: "Finally, we screened two MCL1-knockout single cell clones, and although top synthetic lethal hits such as BCL2L1 and WSB2 were consistently observed in both, one of the two clones shared fewer top hits in common with either the anchor screens or the small molecule, highlighting the challenge of generalization that may emerge when using clonal cell lines."

Also in the lines before this statement authors do acknowledge the differences. As authors also mention, there can be many reasons for these differences. One reason could be that it is the result of long term adaptation of the single cell clones; some genes with a low score in the anchor screens are sensitizing in the "adapted" single cell clones (e.g. PGD), which may be the result of the adaptation. RNASeq comparison or expression level measurements of these genes between a single cell clone and an acute knock-out cell, may help to explain this phenomenon.

We agree that further exploration into these differences is needed to fully understand the differences between the approaches and we are happy to provide some more depth to this point in the discussion. Further experimentation on this subject, however, is outside of the scope of the present study, as performing RNAseq would simply show what differences there are, and would serve as hypotheses for additional mechanistic questions, in other words, another complete study! Rather, we think the data as-presented make for the generalizable conclusion that these three approaches to study gene loss-of-function -- small molecules, single cell clones, and anchor screens -- can identify genes private to one approach.

The differences between PARP guides in olaparib and talazoparib screens in A375 cells remains puzzling. It appears that next to olaparib, PARP1 guides are also unable to induce resistance to the other PARP inhibitors niraparib and veliparib (Supp Figure 13). This raises the question whether these compounds are really inhibiting A375 cells and if so, do they do this through targeting PARP1 or off target mechanisms? Dose-response curves in these cells do not follow the typical sigmoidal shape, in contrast to the HAP1 cells (Supp Figure 12) or HeLa cells (Zimmerman et al.). Are these A375 cells a good model to study these inhibitors? In their response, authors argue that the dose of 250 nM was sufficient to identify other mechanisms of resistance in A375 cells in the screens, such as loss of TP53 and PARG. This argument is true if they are also able to validate these genes in the same way as they did for the 3 different PARP guides.

It is certainly possible that the effect of olaparib on A375 cells is more complex than that in either HeLa cells or HAP1 cells. This could be due to the role of other PARP proteins -- if PARP2, for example, plays a compensatory role in A375 cells, but has a different response to a small molecule inhibitor, then the overall dose-response curve may not be sigmoidal. True off-target effects (i.e. non-PARP protein targeting) of the compounds are also a possibility. Of note, other groups have tested PARP inhibitors on A375 cells. For example, veliparib has been shown to reduce cell viability

and promote pro-apoptotic activity (PMID: 29956724), and olaparib was demonstrated to induce cytostatic and pro-apoptotic effects in A375 cells (PMID: 31185226).

However, we disagree with the reviewer that it is reasonably likely that the resistance phenotype observed with TP53 or PARG observed with both olaparib and talazoparib is speculative unless validated in a focused experiment -- in other words, that these top two hits are false positives in the screen, and a validation experiment is thus required to discern if they are true hits. Both genes are well-supported by prior literature, and the likelihood of two different small molecules having off-target mechanisms that nonetheless share the same resistance mechanism triggered by the loss of the same two genes is vanishingly small. Likewise, on the sensitization side, we still identified enrichment of various DNA damage repair genes (Fig. 5c) in A375 cells.

Were our conclusion that A375 cells are particularly useful model for studying the effect of PARP inhibitors, then we fully agree that the data are unresponsive. That, however, is not our goal in this manuscript. We believe it is important to show all the data for the experiments that we attempted when developing this screening approach, not only those that "look good" and match prior expectations, and we have been careful to avoid any definitive statements about the mechanism of PARP inhibitors based on data acquired only from A375 cells.

We have alerted the reader by noting "Further work will be necessary to understand the mechanistic basis of this difference."

minor comment: I believe the correct word to describe the entry of replication deficient lentiviral vectors in cells is "transduction" and not "infection", infection is generally used for fully replicative competent viruses.

This has been changed in the text

check supp figures 2-7 as some annotations disappeared

Fixed

Please include in the figure legend of Figure 6b the cell line used

Fixed